# Bayesian Optimization through Gaussian Cox Process Models for Spatio-temporal Data

**Yongsheng Mei**[1]**, Mahdi Imani**[2]**, Tian Lan**[1]
[1]The George Washington University
[2]Northeastern Univeristy
{ysmei, tlan}@gwu.edu, m.imani@northeastern.edu

## Abstract

Bayesian optimization (BO) has established itself as a leading strategy for efficiently optimizing expensive-to-evaluate functions. Existing BO methods mostly rely on Gaussian process (GP) surrogate models and are not applicable to (doubly-stochastic) Gaussian Cox processes, where the observation process is modulated by a latent intensity function modeled as a GP. In this paper, we propose a novel maximum *a posteriori* inference of Gaussian Cox processes. It leverages the Laplace approximation and change of kernel technique to transform the problem into a new reproducing kernel Hilbert space, where it becomes more tractable computationally. It enables us to obtain both a functional posterior of the latent intensity function and the covariance of the posterior, thus extending existing works that often focus on specific link functions or estimating the posterior mean. Using the result, we propose a BO framework based on the Gaussian Cox process model and further develop a Nyström approximation for efficient computation. Extensive evaluations on various synthetic and real-world datasets demonstrate significant improvement over state-of-the-art inference solutions for Gaussian Cox processes, as well as effective BO with a wide range of acquisition functions designed through the underlying Gaussian Cox process model.

## 1 Introduction

Bayesian optimization (BO) has emerged as a prevalent sample-efficient scheme for global optimization of expensive multimodal functions. It sequentially samples the space by maximizing an acquisition function defined according to past samples and evaluations. The BO has shown success in various domains, including device tuning (Dalibard et al., 2017), drug design (Griffiths & Hernández-Lobato, 2020), and simulation optimization (Acerbi & Ji, 2017). Existing BO approaches are often built on the Gaussian process (GP) regression model to account for correlation across the continuous samples/search space and provide the prediction of the unknown functions in terms of the mean and covariance. Such models become insufficient when the observation process – e.g., consisting of point events in a Poisson process – is generated from a latent intensity function, which itself is established by a GP through a non-negative link function.

To this end, doubly-stochastic point process models such as Gaussian Cox processes are commonly adopted in analyzing spatio-temporal data in a Bayesian manner and have been successfully applied to many problems in engineering, neuroscience, and finance (Cunningham et al., 2008; Basu & Dassios, 2002). The problem is typically formulated as a maximum *a posteriori* (MAP) inference of the latent intensity function over a compact domain. Its dependence on a functional form of the latent intensity function makes an exact solution intractable due to integrals over infinite-dimensional distributions. Existing works often focus on the inference problem for specific link functions, e.g., sigmoidal link function (Adams et al., 2009; Gunter et al., 2014) and quadratic link function (Lloyd et al., 2015; Walder & Bishop, 2017), or leverage approximation techniques, e.g., variational Bayesian approximation (Aglietti et al., 2019), mean-field approximation (Donner & Opper, 2018) and path integral approximations of GP (Kim, 2021). However, MAP estimation of the latent intensity function alone is not enough to support BO since optimization strategies in BO must be constructed through acquisition functions that quantify the underlying uncertainty of inference

through both posterior mean and covariance. This requires building *a stochastic model of the latent intensity function* to enable BO with such doubly-stochastic models.

In this paper, we consider Gaussian Cox process models with general smooth link functions and formulate a MAP inference of the functional posterior of the latent intensity function and the covariance of the posterior. We show that by leveraging Laplace approximation and utilizing a change of kernel technique, the problem can be transformed into a new reproducing kernel Hilbert space (RKHS) (Flaxman et al., 2017), where a unique solution exists following the representer theorem and becomes more tractable computationally regarding a new set of eigenfunctions in the RKHS. The proposed approach does not rely on variational Bayesian or path integral approximations in previous work (Aglietti et al., 2019; Kim, 2021).

We apply the inference model (of both posterior mean and covariance of the latent intensity function) to propose a novel BO framework and further develop a Nyström approximation when a closed-form kernel expansion is unavailable. The computation is highly efficient since it leverages approximations in previous BO steps to obtain new ones incrementally. To the best of our knowledge, this is the first work on BO using Gaussian Cox process models with kernel transformation in RKHS. It enables us to consider a wide range of acquisition functions, such as detecting peak intensity, idle time, change point, and cumulative arrivals, through the underlying Gaussian Cox process model. Extensive evaluations are conducted using both synthetic functions in the literature (Adams et al., 2009) and real-world spatio-temporal datasets, including DC crime incidents (DC.gov, 2022), 2D neuronal data (Sargolini et al., 2006), and taxi data of Porto (O'Connell et al., 2015). The proposed method shows significant improvement over existing baselines.

The rest of the paper is organized as follows: Section 2 discusses related work and their differences with our approach. Section 3 develops our solution for both posterior mean and covariance, followed by the proposed BO framework with different acquisition function designs using the Gaussian Cox process model. Section 4 presents extensive evaluations on multiple datasets. Finally, Section 5 states our conclusions and underlines potential future work.

## 2 RELATED WORK

**Gaussian Cox Process Models**   Doubly stochastic Gaussian Cox processes connecting the Poisson process with an underlying GP through a link function have been the golden standards in analyzing and modeling spatio-temporal data in many domains (Cunningham et al., 2008; Basu & Dassios, 2002). The MAP intensity estimation problem is much more challenging than estimating a deterministic intensity function (Flaxman et al., 2017), and discretization (Møller et al., 1998; Diggle et al., 2013) is often needed to ensure tractability. Recent works have exploited specific link function structures and considered Markov chain Monte Carlo methods (Adams et al., 2009; Gunter et al., 2014; Nava et al., 2022), mean-field approximations (Donner & Opper, 2018), Langevin dynamics (Mutny & Krause, 2022), sparse variational Bayesian algorithms with link functions like sigmoidal functions (Aglietti et al., 2019) and quadratic functions (Lloyd et al., 2015), as well as transformations into permanental processes (Flaxman et al., 2017; Walder & Bishop, 2017). Recently, path integral formulation (Kim, 2021) has been proposed as a method for effective posterior mean predicative covariance estimations without being constrained by particular link functions. In this work, we proposed a new method for quantifying both posterior mean and covariance of the latent intensity function, which is required to support BO and various acquisition function designs.

**Bayesian Optimization through Acquisition Functions**   Bayesian optimization (BO) can optimize objective functions that are expensive to evaluate, e.g., neural network hyperparameter tuning (Mei et al., 2023; Masum et al., 2021) and drug discovery (Stanton et al., 2022). BO is a sequential approach to finding a global optimum of an unknown objective function $f(\cdot)$: $\boldsymbol{x}^* = \arg\max_{\boldsymbol{x} \in \mathcal{X}} f(\boldsymbol{x})$, where $\mathcal{X} \subset \mathbb{R}^d$ is a compact set. It at each iteration $k$ obtains a new sample $\boldsymbol{x}_{k+1}$ to evaluate $f(\boldsymbol{x}_{k+1})$, update the model of $f$ (often using a posterior GP), and selects the next sample using an acquisition function $a : \mathcal{X} \rightarrow \mathbb{R}$ from the new model, until reaching the optimum $\boldsymbol{x}^*$. Various BO strategies have been proposed, including expected improvement (EI) (Močkus, 1975), knowledge gradient (KG) (Frazier et al., 2009), probability of improvement (PI) (Kushner, 1964), upper-confidence bounds (UCB) (Lai & Robbins, 1985), and entropy search (ES) (Hernández-Lobato et al., 2014). Despite the success of BO in many practical problems, most

Table 1: Common link functions $\kappa(x)$ and their derivatives and inverses.

| | $\kappa(x)$ | $\dot{\kappa}(x)$ | $\ddot{\kappa}(x)$ | $\kappa^{-1}(x)$ |
|---|---|---|---|---|
| Exponential | $\exp(x)$ | $\exp(x)$ | $\exp(x)$ | $\log(x)$ |
| Quadratic | $x^2$ | $2x$ | $2$ | $\sqrt{x}$ |
| Sigmoidal | $(1+\exp(-x))^{-1}$ | $\frac{\exp(-x)}{(1+\exp(-x))^2}$ | $\frac{(1-\exp(x))\exp(x)}{(\exp(x)+1)^3}$ | $-\log(x^{-1}-1)$ |
| Softplus | $\log(1+\exp(x))$ | $(1+\exp(-x))^{-1}$ | $\frac{\exp(-x)}{(1+\exp(-x))^2}$ | $\log(\exp(x)-1)$ |

existing work have not considered BO with Gaussian Cox process models, where point events in Poisson processes are modulated by latent GP intensity functions.

## 3 METHODOLOGY

### 3.1 GAUSSIAN COX PROCESS MODEL

We consider a Lebesgue measurable compact observation space $\mathcal{S} \subset \mathbb{R}^d$ and a latent random function $g(\cdot) : \mathcal{S} \to \mathbb{R}$ following Gaussian Process (GP), denoted by $\mathcal{GP}(g(\boldsymbol{t})|\boldsymbol{\mu}, \boldsymbol{\Sigma})$ with mean and covariance $\theta_{\boldsymbol{\mu},\boldsymbol{\Sigma}} \triangleq (\boldsymbol{\mu}, \boldsymbol{\Sigma})$, respectively. Observations are generated from a point process modulated by the latent random function $g(\boldsymbol{t})$ (through a deterministic, non-negative link function that connects $g(\boldsymbol{t})$ with latent intensity $\lambda(\boldsymbol{t}) = \kappa(g(\boldsymbol{t}))$). Given a set of $n$ observed point events $\{\boldsymbol{t}_i\}_{i=1}^n$ in the observation space $\mathcal{S}$, we consider the likelihood function (in terms of GP parameters $\theta_{\boldsymbol{\mu},\boldsymbol{\Sigma}}$) that is formulated as an expectation over the space of latent random functions, i.e.,

$$p(\{\boldsymbol{t}_i\}_{i=1}^n | \theta_{\boldsymbol{\mu},\boldsymbol{\Sigma}}) = \int_{g(\boldsymbol{t})} p(\{\boldsymbol{t}_i\}_{i=1}^n | g(\boldsymbol{t}))\, p(g(\boldsymbol{t})|\theta_{\boldsymbol{\mu},\boldsymbol{\Sigma}})\, \mathrm{d}g(\boldsymbol{t}), \tag{1}$$

where $p(g(\boldsymbol{t})|\theta_{\boldsymbol{\mu},\boldsymbol{\Sigma}})$ is the Gaussian prior. Within equation (1), the log-probability conditioned on a specific random function $g(\boldsymbol{t})$ takes the following form:

$$\log p(\{\boldsymbol{t}_i\}_{i=1}^n | g(\boldsymbol{t})) = \sum_{i=1}^n \log \lambda(\boldsymbol{t}_i) - \int_{\mathcal{S}} \lambda(\boldsymbol{t})\, \mathrm{d}\boldsymbol{t}, \tag{2}$$

where latent intensity function $\lambda(\boldsymbol{t}) = \kappa(g(\boldsymbol{t}))$ is obtained using the deterministic non-negative link function $\kappa(\cdot) : \mathbb{R} \to \mathbb{R}^+$. For simplicity of notations, we will use a short form $\boldsymbol{g}$ to represent $g(\boldsymbol{t})$ in the rest of the paper. Table 1 shows some common link functions. Existing works on Gaussian Cox process estimation often focus on specific link functions.

To develop a BO framework using Gaussian Cox process models, we consider the problem of estimating both the mean and covariance of the Gaussian Process (GP) posterior. Equation (1) and equation (2) outline the probabilistic surrogate model aligning with this purpose. In the following, we first propose a solution to the posterior mean and covariance estimation problem using Laplace approximation and change of kernel into a new RKHS. Following existing work (Møller et al., 1998; Diggle et al., 2013), we consider a discretization of the observation space to ensure tractability of numerical computation and further propose a Nyström approximation. Then, we will present our BO framework built upon these results, along with new acquisition function designs based on the Gaussian Cox process model.

### 3.2 ESTIMATING POSTERIOR MEAN AND COVARIANCE

Let $f(\boldsymbol{g}) \triangleq p(\{\boldsymbol{t}_i\}_{i=1}^n | \boldsymbol{g}) p(\boldsymbol{g}|\theta_{\boldsymbol{\mu},\boldsymbol{\Sigma}})$ denote the multiplication term inside the integral of equation (1). We apply the Laplace approximation (Illian et al., 2012) to estimate the mean and variance of the Gaussian posterior $p(\boldsymbol{g}|\{\boldsymbol{t}_i\}_{i=1}^n)$, which requires maximizing $\log f(\boldsymbol{g})$. The result using Laplace approximation is provided in the following lemma (proof in Appendix A.).

**Lemma 1** (Laplace approximation). *For the $d$-dimensional multivariate distribution regarding $\boldsymbol{t} \in \mathbb{R}^d$, given a mode $\hat{\boldsymbol{g}}$ such that $\hat{\boldsymbol{g}} = \arg\max_{\boldsymbol{g}} \log f(\boldsymbol{g})$, the likelihood (1) can be approximated as:*

$$\int_{\boldsymbol{g}} f(\boldsymbol{g})\, \mathrm{d}g \approx f(\hat{\boldsymbol{g}}) \frac{(2\pi)^{\frac{d}{2}}}{|\boldsymbol{A}|^{\frac{1}{2}}}, \tag{3}$$

where $\boldsymbol{A} \triangleq -\nabla^2_{\boldsymbol{g}=\hat{\boldsymbol{g}}} \log f(\hat{\boldsymbol{g}})$ *is the Hessian matrix.*

Lemma 1 provides the Laplace approximation to the model posterior $p(\boldsymbol{g}|\{\boldsymbol{t}_i\}_{i=1}^n)$ centered on the MAP estimate problem:

$$p(\boldsymbol{g}|\{\boldsymbol{t}_i\}_{i=1}^n) = \frac{f(\boldsymbol{g})}{\int_{\boldsymbol{g}} f(\boldsymbol{g})\, \mathrm{d}\boldsymbol{g}} = \frac{|\boldsymbol{A}|^{\frac{1}{2}}}{(2\pi)^{\frac{d}{2}}} \exp\left[-\frac{1}{2}(\boldsymbol{g}-\hat{\boldsymbol{g}})^{\mathrm{T}} \boldsymbol{A}(\boldsymbol{g}-\hat{\boldsymbol{g}})\right] \sim \mathcal{N}(\boldsymbol{g}|\hat{\boldsymbol{g}}, \boldsymbol{A}^{-1}), \quad (4)$$

where $\hat{\boldsymbol{g}}$ and $\boldsymbol{A}^{-1}$ are the mean $\boldsymbol{\mu}$ and covariance $\boldsymbol{\Sigma}$ of the GP posterior, respectively.

Before we start solving the posterior mean and covariance, we utilize Lemma 1 to rewrite the Gaussian Cox process log-likelihood in equation (1) to obtain:

$$\log p(\{\boldsymbol{t}_i\}_{i=1}^n|\theta_{\boldsymbol{\mu},\boldsymbol{\Sigma}}) = \log p(\{\boldsymbol{t}_i\}_{i=1}^n|\hat{\boldsymbol{g}}) + \log p(\hat{\boldsymbol{g}}|\theta_{\boldsymbol{\mu},\boldsymbol{\Sigma}}) + \frac{d}{2}\log 2\pi - \frac{1}{2}\log|\boldsymbol{A}|$$
$$\overset{(a)}{\approx} \log p(\{\boldsymbol{t}_i\}_{i=1}^n|\hat{\boldsymbol{g}}) + \text{const.}, \tag{5}$$

where approximation $(a)$ follows from the Bayesian information criterion (BIC) (Schwarz, 1978). BIC further simplifies the Laplace approximation by assuming that when the event number $n$ is large, the prior $p(\boldsymbol{g}|\theta_{\boldsymbol{\mu},\boldsymbol{\Sigma}})$ is independent of $n$ and the term $p(\{\boldsymbol{t}_i\}_{i=1}^n|\hat{\boldsymbol{g}})$ will dominate the rest. It allows us to treat other independent terms as a constant in this inference problem.

We can compute the estimate $\hat{\boldsymbol{g}}$ via a maximization problem $\max_{\boldsymbol{g}} \log f(\boldsymbol{g})$ in Lemma 1, which is equivalent to minimizing the negative log-likelihood derived in equation (5). Based on the likelihood of the Poisson point process, we have:

$$\min_{\boldsymbol{g}} \left\{ -\sum_{i=1}^n \log \kappa(g(\boldsymbol{t}_i)) + \int_{\mathcal{S}} \kappa(g(\boldsymbol{t}))\, \mathrm{d}\boldsymbol{t} \right\}. \tag{6}$$

To tackle the optimization problem in equation (6), we utilize the concept of reproducing kernel Hilbert space (RKHS), which is a Hilbert space of functions $h : \mathcal{S} \to \mathbb{R}$ where point evaluation is a continuous linear functional. Given a non-empty domain $\mathcal{S}$ and a symmetric positive definite kernel $k : \mathcal{S} \times \mathcal{S} \to \mathbb{R}$, a unique RKHS $\mathcal{H}_k$ can be constructed. If we can formulate a regularized empirical risk minimization (ERM) problem as $\min_{h \in \mathcal{H}_k} R(\{h(\boldsymbol{t}_i)\}_{i=1}^n) + \gamma\Omega(\|h(\boldsymbol{t})\|_{\mathcal{H}_k})$, where $R(\cdot)$ denotes the empirical risk of $h$, $\gamma$ is the penalty factor, and $\Omega(\cdot)$ is a non-decreasing error in the RKHS norm, a unique optimal solution exists given by representer theorem (Schölkopf et al., 2001) as $h^*(\cdot) = \sum_{i=1}^n \alpha_i k(\boldsymbol{t}_i, \cdot)$, and the optimization can be cast regarding dual coefficients $\alpha \in \mathbb{R}^n$.

Since $\kappa(\cdot)$ is non-negative smooth link function, we define $h(\boldsymbol{t}) \triangleq \kappa^{\frac{1}{2}}(g(\boldsymbol{t}))$ so that $h^2(\boldsymbol{t}) = \kappa(g(\boldsymbol{t})) : \mathcal{S} \to \mathbb{R}^+$. This definition will provide us with access to the property of $L_2$-norm that simplifies the problem-solving in the next. Then, we can formulate a minimization problem of the penalized negative log-likelihood as regularized ERM, with a penalty factor $\gamma$, given by:

$$\min_{h(\boldsymbol{t})} \left\{ -\sum_{i=1}^n \log h^2(\boldsymbol{t}_i) + \int_{\mathcal{S}} h^2(\boldsymbol{t})\, \mathrm{d}\boldsymbol{t} + \gamma\|h(\boldsymbol{t})\|^2_{\mathcal{H}_k} \right\}. \tag{7}$$

However, equation (7) does not allow a direct application of the representer theorem due to the existence of an extra integral term $\int_{\mathcal{S}} h^2(\boldsymbol{t})\, \mathrm{d}\boldsymbol{t}$ in the optimization. To this end, we show that the term can be merged into the square norm term $\gamma\|h(\boldsymbol{t})\|^2_{\mathcal{H}_k}$ by a change of kernel technique (resulting in a new RKHS) and using the Mercer's theorem (Rasmussen & Williams, 2006). Specifically, note that the integral term actually defines a $L_2$-norm of $h(\boldsymbol{t})$. Taking Mercer's expansion of the integral term and the square norm term in $\mathcal{H}_k$, we can add these two expansions together and view the result as Mercer's expansion of the RKHS square norm regarding a new kernel. It transforms the optimization in equation (7) into a new RKHS (concerning the new kernels), thus enabling the application of the representer theorem. The result is stated in the following lemma (proof in Appendix B).

**Lemma 2** (Kernel transformation). *The minimization objective $J(h)$ in equation (7) can be written using a new kernel $\tilde{k}(\boldsymbol{t}, \boldsymbol{t}')$ as:*

$$J(h) = -\sum_{i=1}^n \log h^2(\boldsymbol{t}_i) + \|h(\boldsymbol{t})\|^2_{\mathcal{H}_{\tilde{k}}}, \tag{8}$$

*where the new kernel function defined by Mercer's theorem is $\tilde{k} = \sum_{i=1}^{\infty} \eta_i(\eta_i + \gamma)^{-1}\phi_i(\boldsymbol{t})\phi_i(\boldsymbol{t}')$ given that $\{\eta_i\}_{i=1}^{\infty}$, $\{\phi_i(\boldsymbol{t})\}_{i=1}^{\infty}$ represent the eigenvalues and orthogonal eigenfunctions of the kernel function $k$, respectively.*

The new objective (8) is now solvable through the application of the representer theorem. Hence, we can derive the solution $\hat{h}(\boldsymbol{t})$ for the original minimization problem in equation (7) and further obtain the $\hat{g}$ to maximize the likelihood (1), which leads to the following theorem (proof in Appendix C).

**Theorem 1** (Posterior mean). *Given observations $\{\boldsymbol{t}_i\}_{i=1}^{n}$ in $\mathcal{S}$, the posterior mean $\boldsymbol{\mu}$ of GP is the solution $\hat{g}$ of the minimization problem (8), taking the form $\hat{g} = \kappa^{-1}(\hat{h}^2(\boldsymbol{t}))$, where $\hat{h}(\cdot) = \sum_{i=1}^{n} \alpha_i \tilde{k}(\boldsymbol{t}_i, \cdot)$.*

Theorem 1 gives the solution of posterior mean estimate using the kernel transformation technique in Lemma 2. Similarly, posterior covariance in equation (4) can be obtained in the next theorem (proof in Appendix D).

**Theorem 2** (Posterior covariance). *Given observations $\{\boldsymbol{t}_i\}_{i=1}^{n}$ in $\mathcal{S}$, the posterior covariance matrix $\boldsymbol{A}^{-1}$ is given by*

$$\boldsymbol{A}^{-1} = \left[ \boldsymbol{\Sigma}^{-1} - \mathrm{diag}\left( \begin{cases} \frac{\ddot{\kappa}(\hat{\boldsymbol{g}}_i)\kappa(\hat{\boldsymbol{g}}_i) - \dot{\kappa}^2(\hat{\boldsymbol{g}}_i)}{\kappa^2(\hat{\boldsymbol{g}}_i)} - \ddot{\kappa}^2(\hat{\boldsymbol{g}}_i)\Delta\boldsymbol{t} & i = j \\ -\ddot{\kappa}^2(\hat{\boldsymbol{g}}_j)\Delta\boldsymbol{t} & i \neq j \end{cases} \right) \right]^{-1}, \tag{9}$$

*where $\hat{\boldsymbol{g}}_i, \hat{\boldsymbol{g}}_j$ represent $\hat{g}(\boldsymbol{t}_i), \hat{g}(\boldsymbol{t}_j)$, and $j$ and $\Delta\boldsymbol{t}$ are from the $m$-partition Riemann sum of the second term in equation (6) as $\int_{\mathcal{S}} \kappa(\hat{\boldsymbol{g}})\,\mathrm{d}\boldsymbol{t} \approx \sum_{j=1}^{m} \kappa(\hat{\boldsymbol{g}}_j)\Delta\boldsymbol{t}$.*

We note that in Theorem 2, the diagonal values of the covariance matrix $\boldsymbol{A}^{-1}$ exhibit two distinct patterns, depending on whether the observation point $\boldsymbol{t}_i$ overlaps with the partition of the Riemann sum $\boldsymbol{t}_j$. Results in this section establish posterior mean and covariance estimates for the Gaussian Cox process model, which lays the foundation for our BO framework.

## 3.3 NUMERICAL KERNEL APPROXIMATION

The computation of posterior mean and covariance requires solving the new kernel function $\tilde{k}$ in Lemma 2, which may not yield a closed-form solution for kernels that cannot be expanded explicitly by Mercer's theorem. To tackle this, we discretize space $\mathcal{S}$ with a uniform grid $\{\boldsymbol{x}_i\}_{i=1}^{m}$, and propose an approximation of the kernel matrix $\boldsymbol{K_{tt}}$ via a $m \times m$ symmetric positive definite Gram matrix $\boldsymbol{K_{xx}} : \mathbb{R}^m \to \mathbb{R}^m$ using Nyström approximation (Baker & Taylor, 1979). This method is highly efficient since it leverages approximations in previous BO steps and obtains the next approximation from new samples in an incremental fashion. We integrate the grid into kernel matrix as:

$$\hat{\boldsymbol{K}}_{\boldsymbol{tt}} = \boldsymbol{K_{tx}}\boldsymbol{K_{xx}}^{-1}\boldsymbol{K_{xt}}, \quad \boldsymbol{K_{xx}} = \boldsymbol{U}\boldsymbol{\Lambda}\boldsymbol{U}^{\mathrm{T}} = \sum_{i=1}^{m} \lambda_i^{\mathrm{mat}}\boldsymbol{u}_i\boldsymbol{u}_i^{\mathrm{T}}, \tag{10}$$

where $\lambda_i^{\mathrm{mat}}, \boldsymbol{u}_i$ are the eigenvalue and eigenvector of $\boldsymbol{K_{xx}}$, and $\boldsymbol{\Lambda} \triangleq \mathrm{diag}(\lambda_1, \ldots, \lambda_m)$ is the diagonal matrix of eigenvalues. The Nyström method provide us with the estimates of the eigenvalue and eigenfunction of the Mercer's expansion, i.e.,

$$\hat{\eta}_i = \frac{1}{m}\lambda_i^{\mathrm{mat}}, \quad \hat{\phi}_i(\boldsymbol{t}) = \frac{\sqrt{m}}{\lambda_i^{\mathrm{mat}}}\boldsymbol{k_{tx}}\boldsymbol{u}_i, \tag{11}$$

where $\boldsymbol{k_{tx}} = (k(\boldsymbol{t}, \boldsymbol{x}_i))_{i=1}^{m}$. This approximation enables efficient computation of posterior mean and covariance using the proposed method while achieving superior estimates than existing baselines as shown later in the evaluation, given by the following lemma (proof in Appendix E). Also, since BO requires step-wise intensity estimation, Nyström method allows iterative approximation based on $\boldsymbol{K_{tt}}$ obtained in previous steps to facilitate the estimation on a larger observation space in the next.

**Lemma 3** (Nyström approximation). *Based on eigenvalue and eigenfunction in equation (11), the new kernel matrix can be approximated as:*

$$\hat{\tilde{\boldsymbol{K}}}_{\boldsymbol{tt}} = \boldsymbol{K_{tx}}\boldsymbol{U}\left( \frac{1}{m}\boldsymbol{\Lambda}^2 + \gamma\boldsymbol{\Lambda} \right)^{-1}\boldsymbol{U}^{\mathrm{T}}\boldsymbol{K_{xt}}, \tag{12}$$

*where $\boldsymbol{K_{tx}}^{\mathrm{T}} = (\boldsymbol{k}_{\boldsymbol{t}_i\boldsymbol{x}})_{i=1}^{n} = \boldsymbol{K_{xt}}$ are $m \times n$ matrices.*

### 3.4 BAYESIAN OPTIMIZATION OVER ESTIMATED INTENSITY

In this section, we introduce our proposed BO framework built upon the Gaussian Cox process model and introduce new acquisition function designs guided by the estimated posterior mean and covariance $\theta_{\mu, \Sigma}$ that provides a stochastic surrogate model over $\mathcal{T} \subset \mathbb{R}^d$. Specifically, in each step $i$, given the current observed region $\mathcal{S}_i \subset \mathcal{T}$, our BO obtains a Gaussian Cox process model using posterior mean and covariance estimates and then samples the next observation (e.g., events in a small interval $\tau$ from unknown parts of $\mathcal{T}$) according to an acquisition function to expand the observations to $\mathcal{S}_{i+1} = \{\tau\} \cup \mathcal{S}_i$. This process starts from an initial $\mathcal{S}_0 \subset \mathcal{T}$ with known events $\boldsymbol{t}_0 \in \mathcal{S}_0$ and continues to update the Gaussian Cox process model (as new samples are iteratively obtained) until the BO terminates. The acquisition function is designed to capture the desired optimization objective (e.g., finding a peak through expected improvement) relying on the proposed Gaussian Cox process model.

The Gaussian Cox process model introduced in this paper enables a wide range of BO acquisition functions. We note that having only the posterior mean estimate is not enough to support BO acquisition functions, which require estimating the uncertainty of the underlying surrogate model, e.g., through the notion of probability of improvement (PI) (Kushner, 1964), upper-confidence bounds (UCB) (Lai & Robbins, 1985), and expected improvement (EI) (Močkus, 1975). We demonstrate the design of four acquisition functions enabled by our model in this section, while the BO framework can be readily generalized to a wide range of BO problems.

**Peak intensity prediction:** This can be achieved by many forms of acquisition functions (such as UCB, PI, and EI) based on our proposed model. We take UCB as an example, i.e.,

$$a_{\text{UCB}}(\boldsymbol{t}; \omega_1) = \mu(\boldsymbol{t}) + \omega_1 \sigma(\boldsymbol{t}), \tag{13}$$

where $\mu$ denotes the posterior mean, $\omega_1$ is the scaling coefficient, and $\sigma$ represents the standard deviation obtained from posterior covariance. This formulation contains explicit exploitation and exploration terms, allowing effective identification of the latent peak in $\mathcal{S}_{\text{peak}}$.

**Maximum idle time:** We can calculate the distribution of the number of arrivals of a Gaussian Cox process model:

**Proposition 1.** *Let the point process $N(\boldsymbol{t})$ be modulated by a latent intensity function $\kappa(\boldsymbol{g})$, then the probability of the number of arrivals in the region $[a, b]$ is given by $\Pr(N(a, b) = n) = (n!)^{-1} \exp(-\int_a^b \kappa(\boldsymbol{g}) \, \mathrm{d}\boldsymbol{t})(\int_a^b \kappa(\boldsymbol{g}) \, \mathrm{d}\boldsymbol{t})^n$.*

We can plug in the posterior mean and variance derived in Theorems 1 and 2 into the above Proposition, i.e., $\hat{\boldsymbol{g}} + \omega_2 \sigma(\boldsymbol{t})$, where $\omega_2$ is the scaling coefficient. Therefore, the acquisition function for maximum idle time detection takes the form:

$$a_{\text{idle}}(\boldsymbol{t}; \epsilon) = \Pr(N(a, b) \leq \epsilon), \tag{14}$$

where $\epsilon$ is a pre-defined small threshold.

**Maximum cumulative arrivals:** Finding an interval with maximum cumulative arrivals can be useful for many data analytic tasks. This acquisition function also differs from peak intensity detection since it considers arrivals within an interval. According to Proposition 1, by replacing $\boldsymbol{g} = \hat{\boldsymbol{g}} + \omega_3 \sigma(\boldsymbol{t})$ with the scaling coefficient $\omega_3$, the cumulative arrival detection acquisition function can be defined as:

$$a_{\text{cum}}(\boldsymbol{t}; \xi) = \Pr(N(a, b) \geq \xi), \tag{15}$$

where $\xi$ is a threshold and can be updated based on samples observed as in existing BO.

**Change point detection:** The occurrence of a sudden change in intensity often indicates a latent incident in the real-world scenario. We can integrate the Bayesian online change point detection algorithm into our framework using Gaussian Cox process models. The following lemma extends the results in (Adams & MacKay, 2007).

**Lemma 4** (Bayesian online change point probability). *Considering the run length $r_k$ with time step $k$ since the last change point given the arrivals so far observed, when $r_k$ reduces to zero, the change point probability is $\Pr(r_k = 0, \boldsymbol{t}_k) = \sum_{r_{k-1}} \Pr(r_{k-1}, \boldsymbol{t}_{k-1}) \pi_{k-1} H(r_{k-1})$, where $\pi$ is the underlying predictive probability and $H(\cdot)$ is the hazard function.*

The estimated posterior mean and variance can be directly plugged into Lemma 4 to obtain a change point acquisition function, i.e.,

$$a_{\text{CPD}}(\boldsymbol{t}; r) = \Pr(r = 0, \boldsymbol{t}). \qquad (16)$$

It is important to note that the posterior mean and variance are re-estimated each time a new sample is obtained. Thus, the change point acquisition function are continuously improved using the knowledge of observed ground-truth arrivals in our BO framework.

The proposed BO framework for intensity estimation with four mentioned acquisition functions are implemented on the real-world dataset. The representative results are provided in Section 4.3.

# 4 EXPERIMENTS

We evaluate our proposed solutions – i.e., Gaussian Cox process inference as well as BO based on the model – using both synthetic functions in the literature (Adams et al., 2009) and real-world spatio-temporal datasets, including DC crime incidents (DC.gov, 2022), 2D neuronal data (Sargolini et al., 2006), and taxi trajectory data in city of Porto (O'Connell et al., 2015). More evaluation results about acquisition function, additional dataset, and sensitivity study, along with further details about evaluation setups are provided in the appendix.

## 4.1 EVALUATION USING SYNTHETIC DATA

### 4.1.1 ESTIMATION OF SYNTHETIC INTENSITY FUNCTIONS

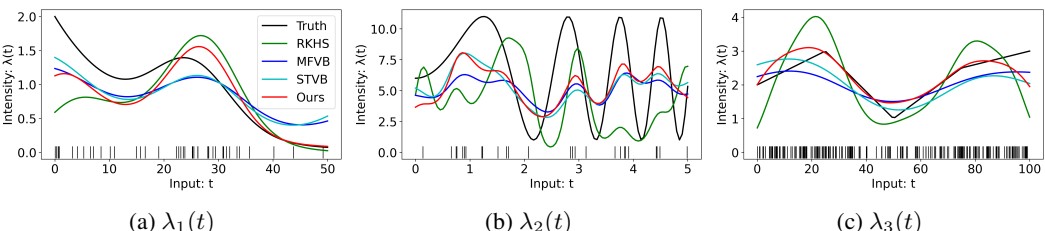

(a) $\lambda_1(t)$          (b) $\lambda_2(t)$          (c) $\lambda_3(t)$

Figure 1: Mean estimations comparison on three types of synthetic data.

Following commonly used examples in the literature (Adams et al. (2009); Kim (2021)) for Gaussian Cox process inference, we consider synthetic intensity functions: $\lambda_1(t) = 2\exp(-t/15) + \exp(-[(t-25)/10]^2)$ for $t \in [0, 50]$, $\lambda_2(t) = 5\sin(t^2) + 6$ for $t \in [0, 5]$, and $\lambda_3$ that is piecewise linear over $(0, 20), (25, 3), (50, 1), (75, 2.5), (100, 3)$ for $t \in [0, 100]$. For three given functions, we generate 44, 27, and 207 synthetic events, respectively. We compare our intensity estimation result with those of existing baselines, including RHKS (Flaxman et al., 2017), MFVB (Donner & Opper, 2018), STVB (Aglietti et al., 2019), and PIF (Kim, 2021). The radial basis function (RBF) kernel and quadratic link function are used in this experiment.

The intensity estimation by different methods is visualized in Fig. 1. Table 2 provides the quantitative results in three metrics, consisting of $l_2$-norm to the ground-truth intensity function and integrated $\rho$-quantile loss ($\text{IQL}_\rho$) (Seeger et al., 2016) [1] where we adopt $\text{IQL}_{.50}$ (mean absolute error) and $\text{IQL}_{.85}$(0.85-quantile). The smaller numbers of these metrics indicate better performances. Since the PIF code has not made available, we borrow the reported numbers in the paper recorded in IQL only. In the table, our method outperforms the baselines in 7 out of all 9 settings. In particular, when considering function $\lambda_2$, our result regarding the $l_2$-norm surpasses the STVB by 1.43. For function $\lambda_3$, the difference between our result and MFVB concerning the $\text{IQL}_{.50}$ is 2.75. This demonstrates the superior performance of the proposed method for Gaussian Cox process inference.

### 4.1.2 BAYESIAN OPTIMIZATION OVER SYNTHETIC INTENSITY

As the central focus of our work, the proposed BO framework with UCB is performed on the Gaussian Cox process model for effective identification of the intensity peaks. The proposed BO frame-

---

[1] $\text{IQL}_\rho \triangleq \int_{\mathcal{S}} 2|\lambda(t_i) - \hat{\lambda}(t_i)|(\rho \text{I}_{\lambda(t_i) > \hat{\lambda}(t_i)} + (1 - \rho)\text{I}_{\lambda(t_i) \le \hat{\lambda}(t_i)}) \, dt$, where I, $\lambda$, and $\hat{\lambda}$ denote the indicator function, ground-truth intensity, and MAP estimation, respectively.

Table 2: Average performance on synthetic data regarding $l_2$-norm and IQL.

| Baselines | $\lambda_1(t)$ | | | $\lambda_2(t)$ | | | $\lambda_3(t)$ | | |
|---|---|---|---|---|---|---|---|---|---|
| | $l_2$ | $IQL_{.50}$ | $IQL_{.85}$ | $l_2$ | $IQL_{.50}$ | $IQL_{.85}$ | $l_2$ | $IQL_{.50}$ | $IQL_{.85}$ |
| Ours | 2.98 | **11.44** | **7.38** | **28.58** | **12.56** | **8.59** | 2.94 | **30.02** | 24.32 |
| RKHS | 4.37 | 16.19 | 12.29 | 44.78 | 18.84 | 11.27 | 6.63 | 54.31 | 53.39 |
| MFVB | 3.15 | 14.36 | 10.88 | 32.60 | 14.41 | 9.40 | 3.82 | 32.77 | **17.89** |
| STVB | **2.96** | 13.64 | 10.26 | 30.01 | 12.86 | 8.66 | 4.19 | 36.68 | 21.86 |
| PIF | – | 12.50 | 9.00 | – | 13.05 | 8.65 | – | 30.81 | 20.03 |

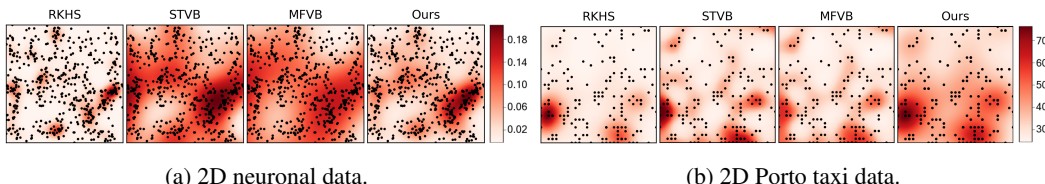

(a) Initial step.  (b) Step 8.  (c) Step 14.  (d) Step 20.

Figure 2: Step-wise visualization of BO on synthetic intensity function.

work facilitates effective intensity estimation and computation of the posterior covariance, enabling the step-wise optimization of next sampling/observation region through UCB acquisition functions, as shown in Fig. 2. The BO process in this test has a budget of 25 steps. As shown in Fig. 2a, the algorithm keeps sampling by maximizing UCB acquisition function and then improving the estimation based on new samples observed. Three peaks are detected at Step 8 (Fig. 2b), Step 14 (Fig. 2c), and Step 20 (Fig. 2d). The completed figure showcasing the consecutive BO procedure is provided in Appendix L. We will present BO with other acquisition functions in Section 4.3.

## 4.2 EVALUATION USING REAL-WORLD DATA

### 4.2.1 INTENSITY ESTIMATION ON 2D SPATIAL DATA

(a) 2D neuronal data.  (b) 2D Porto taxi data.

Figure 3: Mean estimations comparison on two types of 2D real-world data.

We have compared our method with selected baselines on two 2D real-world spatial datasets and provided the qualitative results in Fig. 3. The first dataset is 2D neuronal data in which event locations relate to the position of a mouse moving in an area with recorded cell firing (Sargolini et al., 2006). The other dataset consists of the trajectories of taxi travels in 2013-2014 in Porto, where we consider their starting locations within the coordinates $(41.15, -8.63)$ and $(41.18, -8.60)$.

As depicted in Fig. 3, our method successfully recovered the intensity functions. In Fig. 3a, our approach exhibits an enhanced ability to capture structural patterns reflecting the latent gathering locations of events in comparison to other baselines. Meanwhile, in Fig. 3b, the predicted intensity by our approach is smoother than other baselines, while important structures on the map are all identified. The performance demonstrated in these 2D comparative experiments underscores the validity of our approach for supporting effective BO using the Gaussian Cox models.

### 4.2.2 BAYESIAN OPTIMIZATION ON SPATIO-TEMPORAL DATA

We utilize a public spatio-temporal dataset of crime incidents (i.e., locations and time) in Washington, DC, USA, for 2022. We specifically filtered out 343 crime events involving firearm violence and applied the proposed BO and Gaussian Cox process model to those events. Our initial observations consist of the events that took place on May 20th and June 10th. We employ UCB as the acquisition function in this experiment.

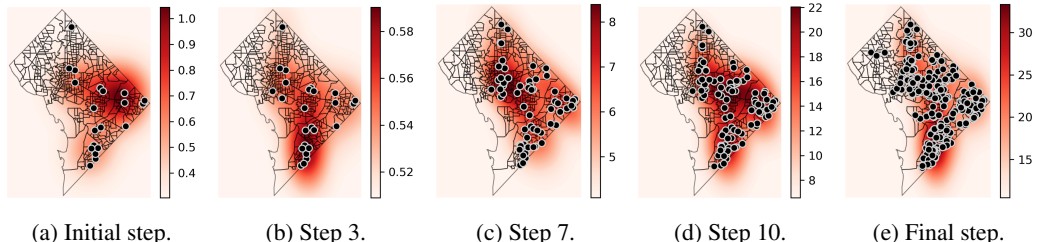

| (a) Initial step. | (b) Step 3. | (c) Step 7. | (d) Step 10. | (e) Final step. |

Figure 4: Step-wise visualization of BO on 2022 DC crime incidents data.

The results are visualized in Fig. 4, where our proposed method successfully identified the general temporal-spatial intensity pattern after 10 steps, observing a small fraction of incidents. Since the BO continues to sample regions with the highest intensity, it keeps exploring and pinpointing primary latent central locations on the map, which are south-east DC area at step 3 (Fig. 4b), downtown area at step 7 (Fig. 4c), and east DC area at step 10 (Fig. 4d). After Step 10, the resulting intensity pattern does not undergo major changes, even as more observations were incorporated, ultimately stabilizing by step 25 (Fig. 4e). The entire BO process of 25 steps took only 91.56 seconds. This demonstrates the effectiveness of our BO framework in efficiently identifying spatial patterns and optimizing unknown functions.

### 4.3 Experiments with Different Acquisition Functions

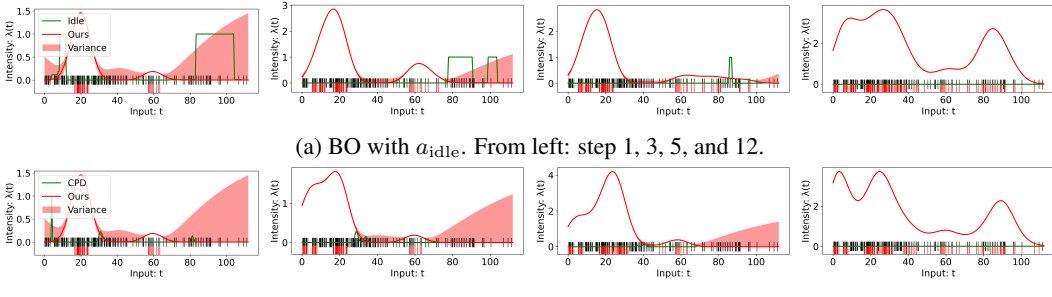

(a) BO with $a_{\mathrm{idle}}$. From left: step 1, 3, 5, and 12.

(b) BO with $a_{\mathrm{CPD}}$. From left: step 1, 2, 4, and 12.

Figure 5: BO with $a_{\mathrm{idle}}$ and $a_{\mathrm{CPD}}$ on coal mining disaster data.

As outlined in Section 3.4, our Gaussian Cox process model and inference solution enable novel acquisition functions beyond the standard ones (e.g., $a_{\mathrm{UCB}}$, $a_{\mathrm{PI}}$, and $a_{\mathrm{EI}}$), to address tasks like idle time and change point detection. We evaluate these novel acquisition functions on a coal mining disaster data (Jarrett, 1979), comprising 190 mining disaster events in the UK with at least 10 casualties recorded between March 15, 1851 and March 22, 1962.

Fig. 5a shows the results for $a_{\mathrm{idle}}$, which prioritizes the exploration of regions with low mean and high variance from the current prediction, implying a higher probability of fewer arrivals (idle time). When $a_{\mathrm{CPD}}$ is adopted, the BO leans toward sampling regions where significant changes in intensity are likely to occur based on the posterior mean and variance, as demonstrated in Fig. 5b. These experiments showcase the versatility of our method in addressing different BO objectives by introducing corresponding acquisition functions.

## 5 Conclusion

In this paper, we provide a novel framework for estimating the posterior mean and covariance of the Gaussian Cox process model by employing Laplace approximation and transforming the problem in a new reproducing kernel Hilbert space. The results enable a novel BO framework based on Gaussian Cox process models, allowing the design of various acquisition functions. Our experimental results on synthetic and real-world datasets demonstrate significant improvement over baselines in estimating spatio-temporal data and supporting BO of unknown functions. The work paves the path to considering more complex scenarios, such as models with the time-variant latent intensity, sparse input spaces, and high dimensions.

ACKNOWLEDGMENTS

This research is based on work supported by the Army Research Office (ARO) under the grant W911NF2110299.

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

## A  PROOF OF LEMMA 1

*Proof.* Assuming a mode $\hat{\boldsymbol{g}}$ such that $\hat{\boldsymbol{g}} = \arg\max_{\boldsymbol{g}} \log f(\boldsymbol{g})$, the condition $\nabla \log f(\hat{\boldsymbol{g}}) = 0$ must be satisfied. Therefore, when expanding $\log f(\boldsymbol{g})$ using Taylor's formula at $\hat{\boldsymbol{g}}$, we have:

$$
\begin{aligned}
\log f(\boldsymbol{g}) &\approx \log f(\hat{\boldsymbol{g}}) + \nabla \log f(\hat{\boldsymbol{g}})(\boldsymbol{g} - \hat{\boldsymbol{g}}) + \frac{1}{2}(\boldsymbol{g} - \hat{\boldsymbol{g}})^{\mathrm{T}} \nabla^2_{\boldsymbol{g}=\hat{g}} \log f(\hat{\boldsymbol{g}})(\boldsymbol{g} - \hat{\boldsymbol{g}}) \\
&= \log f(\hat{\boldsymbol{g}}) + \frac{1}{2}(\boldsymbol{g} - \hat{\boldsymbol{g}})^{\mathrm{T}} \nabla^2_{\boldsymbol{g}=\hat{g}} \log f(\hat{\boldsymbol{g}})(\boldsymbol{g} - \hat{\boldsymbol{g}}).
\end{aligned}
\tag{17}
$$

Let $\boldsymbol{A} \triangleq -\nabla^2_{\boldsymbol{g}=\hat{g}} \log f(\hat{\boldsymbol{g}})$. Based on the derived expansion (17), the likelihood in equation (1) can be approximated as:

$$
\int_{\boldsymbol{g}} f(\boldsymbol{g}) \, \mathrm{d}\boldsymbol{g} \approx f(\hat{\boldsymbol{g}}) \int_{\boldsymbol{g}} \exp\left[-\frac{1}{2}(\boldsymbol{g} - \hat{\boldsymbol{g}})^{\mathrm{T}} \boldsymbol{A}(\boldsymbol{g} - \hat{\boldsymbol{g}})\right] \mathrm{d}\boldsymbol{g} = f(\hat{\boldsymbol{g}}) \frac{(2\pi)^{\frac{d}{2}}}{|\boldsymbol{A}|^{\frac{1}{2}}}
\tag{18}
$$

This concludes the proof. $\qquad\square$

## B  PROOF OF LEMMA 2

*Proof.* With the uniform convergence on $\mathcal{S} \times \mathcal{S}$, the Mercer's representation of the kernel function $k$ is:

$$
k(\boldsymbol{t}, \boldsymbol{t}') = \sum_{i=1}^{\infty} \eta_i \phi_i(\boldsymbol{t}) \phi_i(\boldsymbol{t}'), \quad \boldsymbol{t}, \boldsymbol{t}' \in \mathcal{S},
\tag{19}
$$

where $\{\eta_i\}_{i=1}^{\infty}$ and $\{\phi_i(\boldsymbol{t})\}_{i=1}^{\infty}$ are the corresponding eigenvalues and orthogonal eigenfunctions of kernel function $k$, respectively.

The second integral term in the original objective (7) is the $L_2$-norm, which is:

$$
\int_{\mathcal{S}} h^2(\boldsymbol{t}) \, \mathrm{d}\boldsymbol{t} = \|h(\boldsymbol{t})\|^2_{L_2(\mathcal{S})}.
\tag{20}
$$

Since $h \in \mathcal{H}_k$, we write it as $h = \sum_{i=1}^{\infty} b_i \psi_i$ using Mercer's theorem, leading to $\|h\|^2_{L_2(\mathcal{S})} = \sum_{i=1}^{\infty} b_i^2$ and $\|h\|^2_{\mathcal{H}_k} = \sum_{i=1}^{\infty} b_i^2 \eta_i^{-1}$. Here, $\{b_i\}_{i=1}^{\infty}$ signifies the eigenvalues, $\{\psi_i\}_{i=1}^{\infty}$ represents the orthogonal eigenfunctions of $h$, and $\{\eta_i\}_{i=1}^{\infty}$ corresponds to the eigenvalues of the original kernel function $k$. Therefore, the original objective in equation (7) can be rewritten as:

$$
\begin{aligned}
J(h) &= -\sum_{i=1}^{n} \log h^2(\boldsymbol{t}_i) + \|h(\boldsymbol{t})\|^2_{L_2(\mathcal{S})} + \gamma \|h(\boldsymbol{t})\|^2_{\mathcal{H}_k} \\
&= -\sum_{i=1}^{n} \log h^2(\boldsymbol{t}_i) + \sum_{j=1}^{\infty} b_j^2 + \gamma \sum_{j=1}^{\infty} b_j^2 \eta_j^{-1} \\
&= -\sum_{i=1}^{n} \log h^2(\boldsymbol{t}_i) + \sum_{j=1}^{\infty} b_j^2 (1 + \gamma \eta_j^{-1}) \\
&= -\sum_{i=1}^{n} \log h^2(\boldsymbol{t}_i) + \sum_{j=1}^{\infty} \frac{b_j^2}{\eta_j (\eta_j + \gamma)^{-1}}.
\end{aligned}
\tag{21}
$$

Considering a eigenvalue of a new kernel $\tilde{k}$ satisfying its eigenvalue $\tilde{\eta}_j = \eta_j(\eta_j + \gamma)^{-1}$, the second and third terms in equation (7) can be merged into a single square norm in $\mathcal{H}_{\tilde{k}}$. In this case, compared to the original expansion in equation (19), the Mercer's expansion of the new RHKS kernel is:

$$
\tilde{k}(\boldsymbol{t}, \boldsymbol{t}') = \sum_{i=1}^{\infty} \frac{\eta_i}{\eta_i + \gamma} \phi_i(\boldsymbol{t}) \phi_i(\boldsymbol{t}'), \quad \boldsymbol{t}, \boldsymbol{t}' \in \mathcal{S},
\tag{22}
$$

and thus the simplified objective is:

$$J(h) = -\sum_{i=1}^{n} \log h^2(\boldsymbol{t}_i) + \|h(\boldsymbol{t})\|_{\mathcal{H}_{\tilde{k}}}^2. \tag{23}$$

This concludes the proof. □

## C PROOF OF THEOREM 1

*Proof.* Since $\sum_{j=1}^{\infty} b_j^2 \eta_j^{-1} < \infty$ if and only if $\sum_{j=1}^{\infty} b_j^2 (\eta_j + \gamma) \eta_j^{-1}$, that is $h \in \mathcal{H}_k$ is equivalent to $h \in \mathcal{H}_{\tilde{k}}$, two RHKS spaces then correspond to exactly the same set of functions. Thus, optimization over $\mathcal{H}_k$ equals the optimization over $\mathcal{H}_{\tilde{k}}$. In this case, we can apply the representer theorem to the simplified objective (8) after kernel transformation into $\tilde{k}$.

Given $\{\boldsymbol{t}_i\}_{i=1}^n$, any $h \in \mathcal{H}_{\tilde{k}}$ can be decomposed into a part that lives in the span of the $\tilde{k}$ and a part orthogonal to it, i.e.:

$$h(\cdot) = \sum_{i=1}^{n} \alpha_i \tilde{k}(\boldsymbol{t}_i, \cdot) + v, \tag{24}$$

where, for $\alpha \in \mathbb{R}^n$ and $v \in \mathcal{H}_{\tilde{k}}$, we have:

$$\langle v, \tilde{k}(\boldsymbol{t}_j, \cdot) \rangle = 0, \forall \boldsymbol{t}_j \in \mathcal{S}. \tag{25}$$

The first term in objective (8) is independent of $v$ and only related to $h(\cdot)$. Using equation (25) the reproducing property, application of $h(\cdot)$ to other points $\boldsymbol{t}_j$ yields:

$$\begin{aligned}
h(\boldsymbol{t}_i) &= \langle h, \tilde{k}(\boldsymbol{t}_i, \cdot) \rangle_{\mathcal{H}_{\tilde{k}}} \\
&= \left\langle \sum_{i=1}^{n} \alpha_i \tilde{k}(\boldsymbol{t}_i, \cdot) + v, \tilde{k}(\boldsymbol{t}_i, \cdot) \right\rangle_{\mathcal{H}_{\tilde{k}}} \\
&= \sum_{i=1}^{n} \alpha_i \tilde{k}(\boldsymbol{t}_i, \boldsymbol{t}_j) + \langle v, \tilde{k}(\boldsymbol{t}_j, \cdot) \rangle_{\mathcal{H}_{\tilde{k}}} \\
&= \sum_{i=1}^{n} \alpha_i \tilde{k}(\boldsymbol{t}_i, \boldsymbol{t}_j).
\end{aligned} \tag{26}$$

As for the second term, since the orthogonal property in equation (25), we get:

$$\begin{aligned}
\|h(\boldsymbol{t})\|_{\mathcal{H}_{\tilde{k}}}^2 &= \left\| \sum_{i=1}^{n} \alpha_i \tilde{k}(\boldsymbol{t}_i, \cdot) + v \right\|_{\mathcal{H}_{\tilde{k}}}^2 \\
&= \left\| \sum_{i=1}^{n} \alpha_i \tilde{k}(\boldsymbol{t}_i, \cdot) \right\|_{\mathcal{H}_{\tilde{k}}}^2 + \|v\|_{\mathcal{H}_{\tilde{k}}}^2 \\
&\geq \left\| \sum_{i=1}^{n} \alpha_i \tilde{k}(\boldsymbol{t}_i, \cdot) \right\|_{\mathcal{H}_{\tilde{k}}}^2,
\end{aligned} \tag{27}$$

where, in the last inequality, the equality case occurs if and only if $v = 0$. Setting the $v = 0$ thus does not have any effect on the first term in equation (8), while strictly reducing the second term. Hence, the minimizer must have $v = 0$ and the solution takes the form:

$$\hat{h}(\cdot) = \sum_{i=1}^{n} \alpha_i \tilde{k}(\boldsymbol{t}_i, \cdot). \tag{28}$$

Since we define $h(\boldsymbol{t}) = \kappa^{\frac{1}{2}}(g(\boldsymbol{t}))$, the mean $\boldsymbol{\mu}$ of GP should be:

$$\hat{\boldsymbol{g}} = \kappa^{-1}\left[\left(\sum_{i,j=1}^{n}\alpha_i \tilde{k}(\boldsymbol{t}_i, \boldsymbol{t}_j)\right)^2\right]. \tag{29}$$

The theorem is proven. $\qquad\square$

## D   PROOF OF THEOREM 2

*Proof.* Since $p(\hat{\boldsymbol{g}}|\theta_{\boldsymbol{\mu},\boldsymbol{\Sigma}}) \sim \mathcal{N}(\boldsymbol{\mu}, \boldsymbol{\Sigma})$, we can expand and rewrite log-likelihood of equation (1) as:

$$\begin{aligned}
\log f(\hat{\boldsymbol{g}}) &= \log p(\{\boldsymbol{t}_i\}_{i=1}^n|\hat{\boldsymbol{g}}) + \log p(\hat{\boldsymbol{g}}|\theta_{\boldsymbol{\mu},\boldsymbol{\Sigma}}) \\
&= \log p(\{\boldsymbol{t}_i\}_{i=1}^n|\hat{\boldsymbol{g}}) - \frac{1}{2}\hat{\boldsymbol{g}}^{\mathrm{T}}\boldsymbol{\Sigma}^{-1}\hat{\boldsymbol{g}} - \frac{1}{2}\log|\boldsymbol{\Sigma}|(2\pi)^d,
\end{aligned} \tag{30}$$

and its second-order gradient with respect to $\hat{\boldsymbol{g}}$ is:

$$\nabla_{\hat{\boldsymbol{g}}}^2 \log f(\hat{\boldsymbol{g}}) = \nabla_{\hat{\boldsymbol{g}}}^2 \log p(\{\boldsymbol{t}_i\}_{i=1}^n|\hat{\boldsymbol{g}}) - \boldsymbol{\Sigma}^{-1}. \tag{31}$$

The first term in equation (31) denote the inhomogeneous Poisson process log-likelihood in equation (2). We approximate its integral term with $m$-partition Riemann sum for all dimensions and compute the second-order gradient, resulting:

$$\begin{aligned}
\nabla_{\hat{\boldsymbol{g}}}^2 \log p(\{\boldsymbol{t}_i\}_{i=1}^n|\hat{\boldsymbol{g}}) &\approx \nabla_{\hat{\boldsymbol{g}}}^2\left(\sum_{i=1}^{n}\kappa(g(\boldsymbol{t}_i)) - \sum_{j=1}^{m}\kappa(g(\boldsymbol{t}_j))\Delta\boldsymbol{t}\right) \\
&= \begin{pmatrix} \frac{\partial^2 \log p(\{\boldsymbol{t}_i\}_{i=1}^n|\hat{\boldsymbol{g}})}{\partial\boldsymbol{g}_1\,\partial\boldsymbol{g}_1} & \cdots & \frac{\partial^2 \log p(\{\boldsymbol{t}_i\}_{i=1}^n|\hat{\boldsymbol{g}})}{\partial\boldsymbol{g}_m\,\partial\boldsymbol{g}_1} \\ \vdots & \ddots & \vdots \\ \frac{\partial^2 \log p(\{\boldsymbol{t}_i\}_{i=1}^n|\hat{\boldsymbol{g}})}{\partial\boldsymbol{g}_1\,\partial\boldsymbol{g}_m} & \cdots & \frac{\partial^2 \log p(\{\boldsymbol{t}_i\}_{i=1}^n|\hat{\boldsymbol{g}})}{\partial\boldsymbol{g}_m\,\partial\boldsymbol{g}_m} \end{pmatrix} \\
&= \mathrm{diag}\left(\begin{cases} \frac{\ddot{\kappa}(\hat{\boldsymbol{g}}_i)\kappa(\hat{\boldsymbol{g}}_i) - \dot{\kappa}^2(\hat{\boldsymbol{g}}_i)}{\kappa^2(\hat{\boldsymbol{g}}_i)} - \ddot{\kappa}^2(\hat{\boldsymbol{g}}_i)\Delta\boldsymbol{t} & i = j \\ -\ddot{\kappa}^2(\hat{\boldsymbol{g}}_j)\Delta\boldsymbol{t} & i \neq j \end{cases}\right).
\end{aligned} \tag{32}$$

Given $\boldsymbol{A} = -\nabla_{\boldsymbol{g}=\hat{\boldsymbol{g}}}^2 \log f(\hat{\boldsymbol{g}})$, we conclude the proof by providing the variance $\boldsymbol{A}^{-1}$ in equation (4) with respect to Gaussian mean $\hat{\boldsymbol{g}}$ by applying a subtraction:

$$\boldsymbol{A}^{-1} = \left(\boldsymbol{\Sigma}^{-1} - \nabla_{\hat{\boldsymbol{g}}}^2 \log p(\{\boldsymbol{t}_i\}_{i=1}^n|\hat{\boldsymbol{g}})\right)^{-1}, \tag{33}$$

where the result of $\nabla_{\hat{\boldsymbol{g}}}^2 \log p(\{\boldsymbol{t}_i\}_{i=1}^n|\hat{\boldsymbol{g}})$ is derived in equation (32). To align the dimension between the covariance and gradient results, we discretize each dimension of the covariance matrix via a size $m$ uniform grids to ensure a lossless computation, where we have $\boldsymbol{\Sigma} = \boldsymbol{\Sigma}_1 \otimes \cdots \otimes \boldsymbol{\Sigma}_d$ with $\otimes$ denoting Kronecker product.

It is worth noting that $\boldsymbol{t}_i$ and $\boldsymbol{t}_j$ may not always be exactly identical in practical implementations. To address this, we can consider them as the same if their absolute difference falls within a small error range. $\qquad\square$

## E   PROOF OF LEMMA 3

*Proof.* We use the $i$th estimated eigenfunction $\hat{\phi}_i(\boldsymbol{t})$ and eigenvalue $\hat{\eta}_i$ to compute the approximation of the new kernel function in Lemma 2 as:

$$\begin{aligned}
\hat{\tilde{k}}(\boldsymbol{t}, \boldsymbol{t}') &= \sum_{i=1}^{m}\frac{\frac{1}{m}\lambda_i^{\mathrm{mat}}}{\frac{1}{m}\lambda_i^{\mathrm{mat}} + \gamma}\hat{\phi}_i(\boldsymbol{t})\hat{\phi}_i(\boldsymbol{t}') \\
&= \sum_{i=1}^{m}\frac{\lambda_i^{\mathrm{mat}}}{\lambda_i^{\mathrm{mat}} + m\gamma}\frac{m}{(\lambda_i^{\mathrm{mat}})^2}\boldsymbol{k}_{\boldsymbol{tx}}\boldsymbol{u}_i\boldsymbol{u}_i^{\mathrm{T}}\boldsymbol{k}_{\boldsymbol{t'x}}^{\mathrm{T}} \\
&= \boldsymbol{k}_{\boldsymbol{tx}}\left[\sum_{i=1}^{m}\frac{m}{(\lambda_i^{\mathrm{mat}} + m\gamma)\lambda_i^{\mathrm{mat}}}\boldsymbol{u}_i\boldsymbol{u}_i^{\mathrm{T}}\right]\boldsymbol{k}_{\boldsymbol{t'x}}^{\mathrm{T}}.
\end{aligned} \tag{34}$$

Since the $\mathbf{\Lambda}$ in equation (10) is the diagonal matrix of every $i$th eigenvalue $\lambda_i^{\mathrm{mat}}$, by applying the derived kernel function estimation (34) to all input observations, the estimation of the new kernel matrix $\tilde{\mathbf{K}}_{\mathbf{tt}}$ becomes:

$$
\begin{aligned}
\hat{\tilde{\mathbf{K}}}_{\mathbf{tt}} &= \mathbf{K}_{\mathbf{tx}} \left[ \sum_{i=1}^{m} (m^{-1}(\lambda_i^{\mathrm{mat}})^2 + \gamma \lambda_i^{\mathrm{mat}})^{-1} \mathbf{u}_i \mathbf{u}_i^{\mathrm{T}} \right] \mathbf{K}_{\mathbf{xt}} \\
&= \mathbf{K}_{\mathbf{tx}} \mathbf{U} \left( \frac{1}{m} \mathbf{\Lambda}^2 + \gamma \mathbf{\Lambda} \right)^{-1} \mathbf{U}^{\mathrm{T}} \mathbf{K}_{\mathbf{xt}},
\end{aligned}
\tag{35}
$$

which concludes the proof. □

## F  PROPOSED ALGORITHMS

We summarize the workflow of the proposed BO framework on Gaussian Cox Process Models in Algorithm 1. The numerical approximation approach for posterior mean and variance estimation is provided in Algorithm 2.

---
**Algorithm 1** BO on the Gaussian Cox Process
---
1: **initialize** Starting observation region $\mathcal{S}_0$, acquisition function $a(\mathbf{t})$, total step $T$
2: **for** $i = 0 : (T - 1)$ **do**
3:  Estimate posterior mean and covariance of the Gaussian Cox process $\theta_{\boldsymbol{\mu}, \boldsymbol{\Sigma}}$ using Theorem 1 and 2 based on observations in region $\mathcal{S}_i$
4:  Find observations in next sampling region $\{\tau\}$ through the acquisition function $a(\mathbf{t}; \theta_{\boldsymbol{\mu}, \boldsymbol{\Sigma}})$ regarding current mean and covariance
5:  $\mathcal{S}_{i+1} \leftarrow \{\tau\} \cup \mathcal{S}_i$
6: **end for**
---

---
**Algorithm 2** Estimations using Nyström Approximation
---
1: **initialize** Kernel matrix $\mathbf{K}$, observed events $\{\mathbf{t}_i\}_{i=1}^n$, discretized grid $\{\mathbf{x}_i\}_{i=1}^m$, random dual coefficient $\alpha_0$, learning rate $\delta$
2: Perform eigendecomposition on grid $\mathbf{K}_{\mathbf{xx}} = \mathbf{U} \mathbf{\Lambda} \mathbf{U}^{\mathrm{T}} = \sum_{i=1}^m \lambda_i^{\mathrm{mat}} \mathbf{u}_i \mathbf{u}_i^{\mathrm{T}}$
3: Derive the eigenvalue and eigenfunction of $\tilde{k}(\mathbf{t}_i, \mathbf{t}_j)$ using equation (11)
4: Update dual coefficient $\alpha$ for new kernel function $\tilde{k}$ using gradient descent:
5: Compute the gradient of optimization objective $J$ in equation (8) as $\nabla_\alpha J$
6: **for** $k = 0 : (n - 1)$ **do**
7:  $\alpha_{k+1} \leftarrow \alpha_k - \delta \nabla_\alpha J$
8: **end for**
9: Compute new kernel matrix $\hat{\tilde{\mathbf{K}}}_{\mathbf{tt}}$ regarding $\tilde{k}$ based on Lemma 3
10: Compute the posterior mean $\hat{\mathbf{g}}$ and covariance $\mathbf{A}^{-1}$, respectively
---

## G  DETAILED EVALUATION SETUPS

We conducted our experiments on the Ubuntu 20.04 system, with Intel(R) Core(TM) i7-6700 4-core CPU (3.4 GHz) and 16.0 GB RAM. The algorithm is implemented in Python 3.8, using main Python libraries NumPy 1.22.3 and Pandas 2.0.3. The code has been made available on GitHub via https://github.com/ysmei97/gaussian_cox_bo.

To evaluate the BO over synthetic intensity in Section 4.1.2, a random ground-truth intensity function is generated (black curve in Fig. 2), where the samples are shown by thinned events (Lewis & Shedler, 1979) in the bottom of plots, marked by black vertical bars. In the experiment, we initialize the region centers $t = (25, 60)$ in the time domain of $[0, 100]$ with a region radius of 2, i.e., region size of 4. Observations in selected regions are highlighted in red vertical bars. We use the UCB acquisition function for identifying intensity peaks. To expedite the BO process, we set the acquisition

values of explored regions as zero to prevent the algorithm from becoming trapped within the same region.

For different acquisitions experiments in Section 4.3, we utilize coal mining disaster data (Jarrett, 1979), comprising 190 mining disaster events in the UK with at least 10 casualties recorded between March 15, 1851 and March 22, 1962. We conducted the experiments with a total budget of 12 steps. The initial region centers are $(20, 60)$, and the region radius is 4. For visualization purposes, we let the year 1851 be the starting point (i.e., 0) of the time axis, shown in Fig. 5.

# H   ADDITIONAL EXPERIMENTS ON ACQUISITION FUNCTIONS

## H.1   INTENSITY PEAK PREDICTION

In Table 3, we provide measures of the $l_2$ distance between our predictions and the ground-truth intensity at the steps where intensity maxima are identified. All three intensity peaks in Fig. 2 (from left to right) correspond to the respective maxima in the table. We include information about the step at which each maximum is identified, along with its current $l_2$ norm, considering different acquisition functions for each maximum. We adopt common acquisition functions for deciding the optimum solution, i.e., UCB, PI, and EI. These results illustrate that all three acquisition functions can effectively identify the peaks in the synthetic intensity. Furthermore, we present the $l_2$ norm every 8 steps, demonstrating a decreasing trend in the distances between the ground truth and predicted intensities over time. This experiment underscores the capability of our method to efficiently locate peaks in a given point process when equipped with an appropriate acquisition function.

Table 3: Step-wise evaluation to intensity peak regarding different acquisition functions (AFs).

| AFs | 1: 1st maximum | | 2: 2nd maximum | | 3: 3rd maximum | | $l_2$-norm every 8 steps | | |
|---|---|---|---|---|---|---|---|---|---|
| | Step | $l_2$ | Step | $l_2$ | Step | $l_2$ | Step 8 | Step 16 | Step 24 |
| UCB | 8 | 10.79 | 14 | 8.91 | 20 | 9.16 | 10.79 | 10.21 | 9.27 |
| EI | 12 | 13.68 | 23 | 8.07 | 8 | 9.92 | 9.92 | 10.54 | 8.51 |
| PI | 9 | 17.01 | 12 | 14.50 | 25 | 11.17 | 17.34 | 11.79 | 11.15 |

## H.2   CUMULATIVE ARRIVAL DETECTION

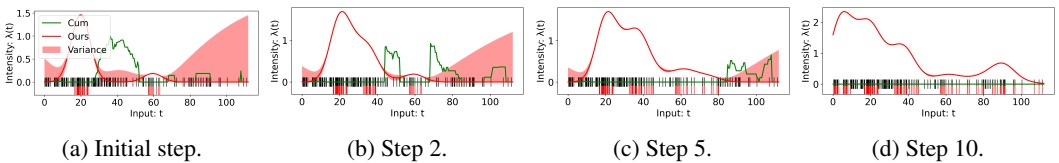

(a) Initial step.  (b) Step 2.  (c) Step 5.  (d) Step 10.

Figure 6: BO with $a_{\mathrm{cum}}$ on coal mining disaster data.

Fig. 6 shows the results for $a_{\mathrm{cum}}$, which indicates the region that contains the most cumulative arrivals. This acquisition function differs from peak intensity detection since the selected region by the former might not necessarily have the global maxima but most point events. As showcased in the figure, the designed acquisition function $a_{\mathrm{cum}}$ can successfully pinpoint the region of interest where more arrivals are possibly gathered.

# I   BAYESIAN OPTIMIZATION ON 2022 USA TORNADO DATA

We utilize a public spatio-temporal dataset of tornado disasters (i.e., locations and time) in the USA in 2022. We specifically filtered out 1126 tornado events with actual loss and applied the proposed BO and Gaussian Cox process model on those events with a budget of 20 steps. We employ the UCB acquisition function in this experiment. The results are visualized in Fig. 7, where our proposed method successfully identified the general temporal-spatial intensity pattern after 6 steps. After Step 12, the resulting intensity pattern does not change significantly as more observations fall in

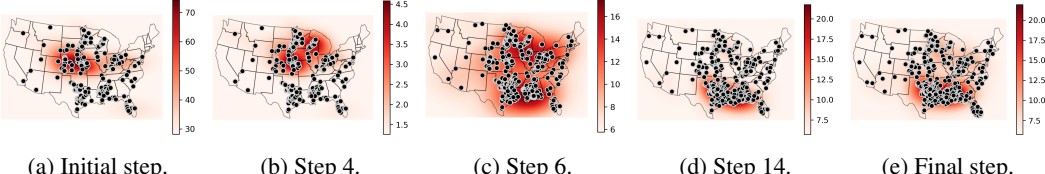

| (a) Initial step. | (b) Step 4. | (c) Step 6. | (d) Step 14. | (e) Final step. |

Figure 7: Step-wise visualization of BO on 2022 USA tornado data.

Table 4: Sensitivity results regarding different link functions in $l_2$-norm, IQL, and errors.

| Baselines | $\lambda_1(t)$ | | | $\lambda_2(t)$ | | | $\lambda_3(t)$ | | |
|---|---|---|---|---|---|---|---|---|---|
| | $l_2$ | $\text{IQL}_{.50}$ | $\text{IQL}_{.85}$ | $l_2$ | $\text{IQL}_{.50}$ | $\text{IQL}_{.85}$ | $l_2$ | $\text{IQL}_{.50}$ | $\text{IQL}_{.85}$ |
| Ours (q) | 2.98 | **11.44** | 7.38 | **28.58** | **12.56** | **8.59** | 2.94 | **30.02** | 24.32 |
| | (1.32) | (2.87) | (3.01) | (7.64) | (1.01) | (2.34) | (0.98) | (5.73) | (3.04) |
| Ours (e) | 3.11 | 12.46 | 8.56 | 31.22 | 15.63 | 12.84 | 4.00 | 31.98 | 26.41 |
| | (1.01) | (3.44) | (3.55) | (8.14) | (1.73) | (2.11) | (1.25) | (6.54) | (6.03) |
| Ours (s) | **2.91** | 11.98 | **7.21** | 29.73 | 15.32 | 11.37 | 4.17 | 33.77 | 24.15 |
| | (1.27) | (3.21) | (3.18) | (7.87) | (1.69) | (2.85) | (1.88) | (7.91) | (5.34) |
| RKHS | 4.37 | 16.19 | 12.29 | 44.78 | 18.84 | 11.27 | 6.63 | 54.31 | 53.39 |
| | (1.64) | (3.21) | (3.04) | (11.56) | (2.81) | (2.34) | (2.78) | (8.20) | (7.89) |
| MFVB | 3.15 | 14.36 | 10.88 | 32.60 | 14.41 | 9.40 | 3.82 | 32.77 | **17.89** |
| | (1.23) | (2.84) | (2.98) | (9.01) | (2.45) | (1.89) | (2.31) | (5.94) | (3.01) |
| STVB | 2.96 | 13.64 | 10.26 | 30.01 | 12.86 | 8.66 | 4.19 | 36.68 | 21.86 |
| | (1.43) | (2.91) | (2.94) | (8.57) | (2.30) | (2.41) | (2.01) | (6.22) | (3.16) |
| PIF (q) | – | 12.50 | 9.00 | – | 13.05 | 8.65 | – | 30.81 | 20.03 |
| | | (4.40) | (4.29) | | (1.88) | (1.70) | | (5.89) | (4.12) |
| PIF (e) | – | 12.29 | 9.30 | – | 15.68 | 12.35 | – | 32.24 | 21.05 |
| | | (4.33) | (3.74) | | (1.67) | (1.99) | | (7.14) | (4.79) |
| PIF (s) | – | 11.58 | 7.62 | – | 14.46 | 10.96 | – | 32.59 | 20.21 |
| | | (3.26) | (3.22) | | (0.83) | (3.20) | | (7.24) | (6.94) |

the predicted high-intensity region, ultimately stabilizing by step 20 (Fig. 7e). The evolution of the intensity of this experiment shows that the southern USA will experience more tornado disasters, demonstrating the effectiveness of our BO framework.

## J SENSITIVITY EXPERIMENTS REGARDING DIFFERENT LINK FUNCTIONS

We change the link functions used in our framework and show the sensitivity results in estimating the synthetic intensity function Table 4. In this table, we use the abbreviations q, e, and s to represent the quadratic, exponential, and softplus link functions, respectively. Detailed results of baseline methods, including statistical errors, are also given in the table. Notably, our results consistently outperform the other baseline methods in 8 out of 9 cases. In particular, for the synthetic function $\lambda_1$, our approach with the softplus link function yields superior results, while the quadratic link function proves effective in handling other scenarios.

## K RUNTIME EXPERIMENTS

We conducted runtime tests of our BO framework on three distinct datasets, each with a trial budget of 20 steps. The step-wise and total runtime, along with the error in parenthesis for each trial, are presented in Table 5. Notably, the test involving 1126 events from the USA tornado dataset requires the most time to complete, whereas the tests on DC crime data with 343 events and coal mining disaster data with 190 events need comparatively less time. Additionally, the step-wise runtime exhibits a noticeable increasing trend as the number of observed events grows over time. This is due to the corresponding increase in estimation complexity as more point events are incorporated into the analysis.

Table 5: BO runtime evaluation (in second) for 20 steps regarding different datasets.

| Datasets | Step 5 time | Step 10 time | Step 15 time | Step 20 time | Total time |
|---|---|---|---|---|---|
| DC crimes | 1.42 (0.56) | 2.55 (0.44) | 4.22 (0.84) | 5.95 (1.28) | 68.61 (1.45) |
| USA tornadoes | 14.71 (2.56) | 58.80 (8.73) | 77.89 (9.75) | 194.23 (14.67) | 820.56 (21.40) |
| Coal mining disasters | 0.57 (0.21) | 0.96 (0.26) | 1.45 (0.32) | 2.71 (0.51) | 25.96 (0.33) |

## L  VISUALIZATION OF THE BAYESIAN OPTIMIZATION PROCEDURE

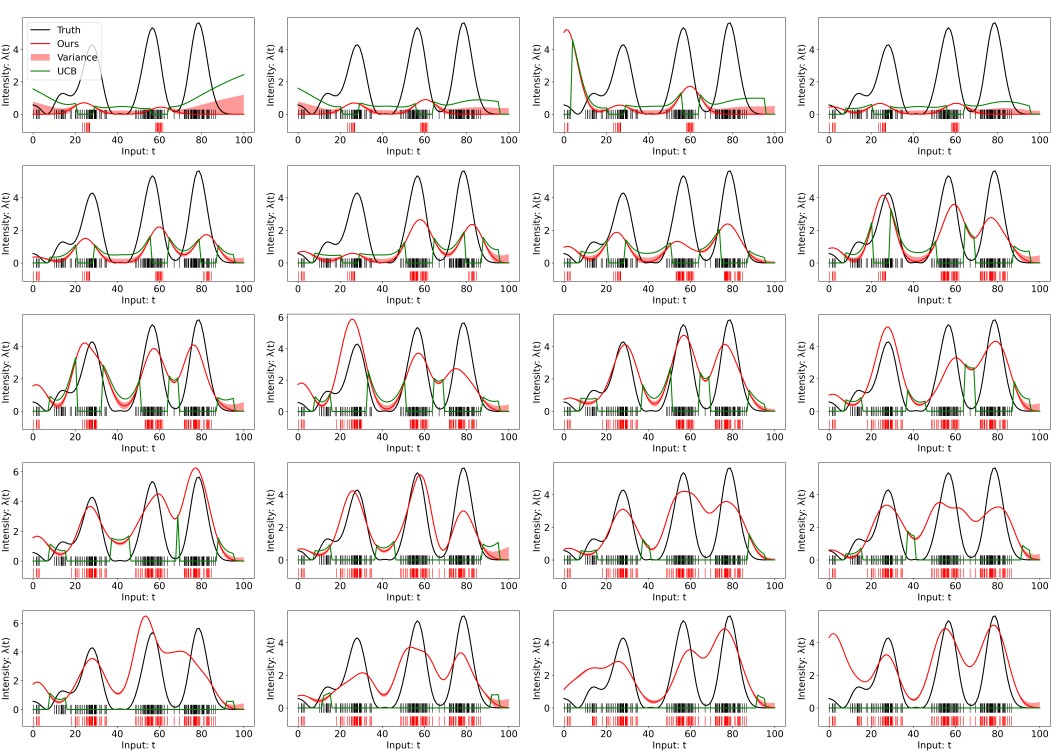

Figure 8: Complete BO procedure from step 1 (top left) to 20 (bottom right) on a synthetic intensity.

In this section, we provide a complete visualization of Fig. 2 in Section 4.1.2. In the figure, the acquisition function depicted by the green curve anticipates the region of interest for the next step based on our posterior estimates of the current step. For instance, in step 1, the acquisition function suggests the region centered at 100. Thus, the algorithm will observe the indicated region at the step 2, though no events exist in that region.

## M  SENSITIVITY EXPERIMENTS REGARDING WEIGHTS IN ACQUISITION FUNCTIONS

Introduced in Section 3.4, $\omega_1, \omega_2, \omega_3$ are weights for mean and covariance terms to balance their contributions in the acquisition functions $a_{\mathrm{UCB}}$, $a_{\mathrm{idle}}$, and $a_{\mathrm{cum}}$, respectively. When these hyperparameter values are small, BO will place higher emphasis on the mean, favoring exploitation in the process. In contrast, when they are large, BO favors exploration by placing higher emphasis on large variance. In our previous experiments, these hyperparameters were fixed at 0.8. In this section, we conducted the sensitivity experiments regarding $\omega_3$ as the illustrative example. Comparing the results in Fig. 9 and Fig. 6 (where $\omega_3 = 0.8$), when we reduce the value of $\omega_3$ to 0.6, the unexplored

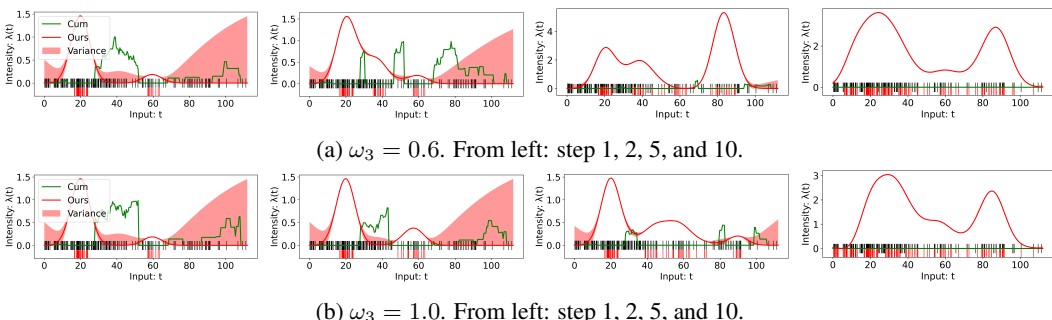

(a) $\omega_3 = 0.6$. From left: step 1, 2, 5, and 10.

(b) $\omega_3 = 1.0$. From left: step 1, 2, 5, and 10.

Figure 9: Sensitivity study about tuning $\omega_3$ for $a_{\text{cum}}$.

areas with high mean and low uncertainty will be prioritized, such as region centered at $t = 30$ in step 2, Fig 9a. Conversely, increasing $\omega_3$ to 1.0, as shown in Fig. 9b, shifts the focus to areas with high uncertainty.

## N  BAYESIAN OPTIMIZATION ON SPARSE SYNTHETIC DATA

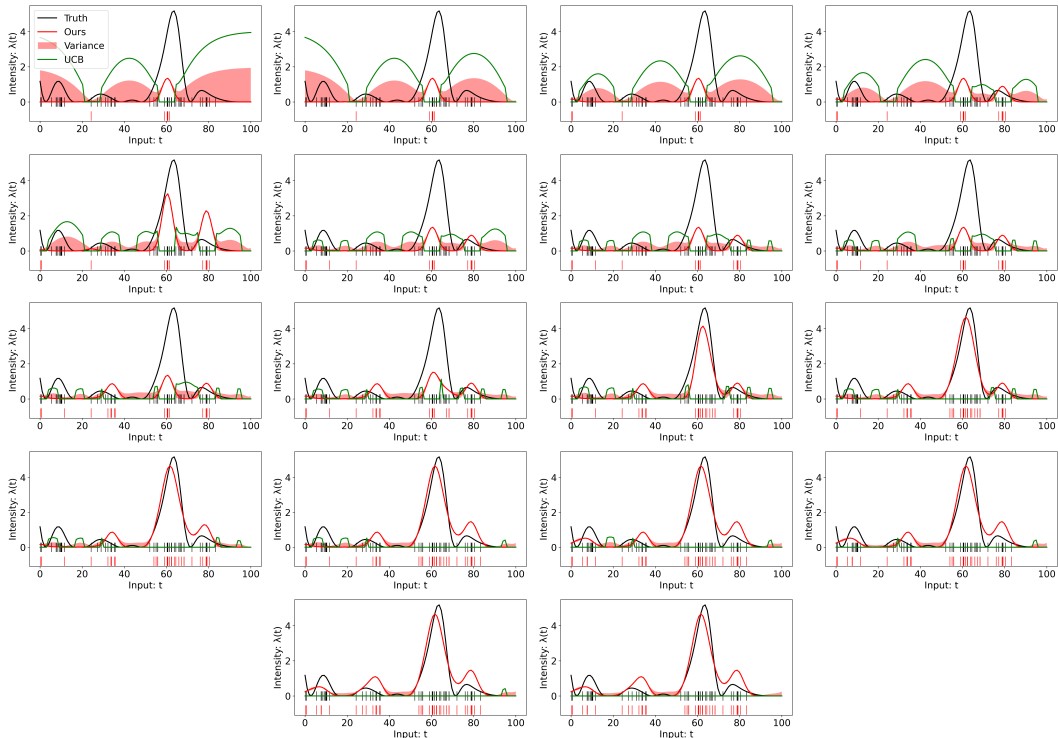

Figure 10: Complete BO procedure from step 1 (top left) to 18 (bottom right) on sparse data.

In Section 3.2, we utilized an approximation (5) assuming the number of events $n$ being large. Therefore, we experiment over a sparse synthetic data with 48 events to demonstrate the capability of the proposed framework. The results of 18 steps are shown in Fig. 10. Initially, the estimation deviates significantly from the ground truth due to the limited observations. However, as the number of observations accumulates during the BO process, the approximation becomes more accurate after each step. In the figure, the method demonstrates its ability to predict latent intensity peaks with relatively sparse data.

(a) $\lambda_1$. (b) $\lambda_2$. (c) $\lambda_3$.

Figure 11: Visualization of tabular results regarding $l_2$-norm.

## O    VISUALIZATION OF TABULAR SYNTHETIC INTENSITY RESULTS

In this section, we visualize the tabular results using box plots for $l_2$-norm metric with three synthetic functions in Section 4.1.1. In the experiments, we adopt ten replicates for each available baseline. The results are provided in Fig. 11, where the red horizontal lines show the medians.

