# OpenReview forum: "Bayesian Optimization through Gaussian Cox Process Models for Spatio-temporal Data"
_ICLR.cc/2024/Conference — ICLR 2024 poster_

### Official Review · Reviewer_dgpT · 2023-10-25

**Soundness:** 3 good
**Presentation:** 4 excellent
**Contribution:** 3 good
**Rating:** 8
**Confidence:** 4

**Summary:**

The paper seeks to build a novel Bayesian Optimization approach that is based on Gaussian Cox Processes for spatio-temporal data. The authors emphasize that Gaussian Cox Processes have never been used within BO settings. The posterior distribution is computed using Laplace approximation and a change of kernel that enables to transform the inference problem into a kernel regression problem. The latter kernel is computed using Nyström approximation. Then the authors present several acquisition functions that are built using posterior mean and posterior variance of the Gaussian Cox Process. Finally the authors illustrate the efficiency of their approach on two types of experiments. First, they show the quality of the mean estimation on synthetic data (4.1.1) and real word data (4.2.1). Second, they show how the BO with UCB function performs for some synthetic dataset (4.1.2) and real world dataset (4.2.2).

**Strengths:**

The paper is well written and the overall approach is scientifically sound and compelling.
The motivation is clear, which is to provide a BO framework with Gaussian Cox processes.
Although Laplace approximation is a standard tool, the authors provide an elegant way to derive the maximum of the log likelihood through a trick that transform the problem into a standard kernel regression problem so that they can use the representer theorem. The main results are clearly detailed. The use of Nystrom approximation is standard, but Lemma 3 helps the reader to figure out what has been implemented.
The numerical experiments section show that the authors made great effort to compare all the components of their methodology with some of the state-of-the-art approaches. In addition, this has been done for simulated data and real-world data.

**Weaknesses:**

Although the paper is well written and has many interesting components, there are a couple of points that need to be detailed.

Major comments:

- My first question is general. In a standard BO setting, we aim at minimizing a function which is costly to evaluate. It seems that this is not the objective of the presented BO problem. It would have been good to make a clear distinction between the two problems.

 - In the literature review, the authors write that "existing works mostly concentrate on the mean estimation". I am not an expert in Cox models, but it seems that this has been investigated in [1]. This means that this approach could have also been tested in the numerical experiments with Bayesian Optimization. It also seems that this cited paper uses some tools (Laplace approximation, eigenfunctions decomposition) similar to the ones in the paper. Would it be possible to highlight the main modeling differences with this paper?
What is the theoretical/computational benefit of the paper's approach compared to the cited paper?

-  In Equation (5), the authors claim that the rest of the terms of the likelihood are dominated by the first term when $n$ is large. However, this assumption is not always true in a BO setting (it is not in the numerical experiments). Could the authors comment that point and provide more details?

- The authors claim that they can use the Nystrom approximation to compute the next approximation from new samples in an incremental fashion. Is this step used in the experiment? How does it work in practice?

Minor comments:

- After reading the proof of Lemma 2, there is a point I did not understand. Why does the function $h$ belong to $\mathcal{H}$? This fact does not look straightforward to me. Perhaps I missed something in the development.

- In the numerical experiments, the authors do not report the results of PIF in Table 2. Is there a reason for that? I've not found it in the paper.

- In figure 2c, it looks like the maximum of the acquisition function is around t=40. This means that we have to sample around $t=40$. However I don't see any sample around $t=40$ in figure 2d.


[1] Hideaki Kim. Fast Bayesian Inference for Gaussian Cox Processes via Path Integral Formulation. Advances in Neural Information Processing Systems  2021.

**Questions:**

See questions above.
Is the code publicly available?
Depending on author responses, I would change my score.

---

> ### Author Response · Authors · 2023-11-16
> **Responses to Reviewer dgpT (1/2)**
>
> Thank you for your constructive suggestions. Our answers to your questions are provided below.
>
> >Reviewer: In a standard BO setting, we aim at minimizing a function which is costly to evaluate. It seems that this is not the objective of the presented BO problem. It would have been good to make a clear distinction between the two problems.
>
>
> ***Response***: Thank you for the question. We would like to clarify that this is indeed our objective. However, expensive-to-evaluate functions relating to discrete point observations over time and space, such as arrivals, events, and occurrences, must be modeled as a Gaussian Cox process, which is a Poisson process modulated by a latent Gaussian process. Standard BO relies on a Gaussian prior and cannot model such expensive-to-evaluate functions with respect to point observation and spatio-temporal data. It requires using doubly stochastic process models, i.e., Gaussian Cox process models considered in this paper.
>
> We clarify that the paper makes two related contributions: (1) a Gaussian Cox process model for expensive-to-evaluate functions relating to discrete point observations over time and space, and (2) a BO framework that builds on such models.
>
> We will also add two illustrative examples to demonstrate the unique advantages of using Gaussian Cox process models for BO. (1) Consider a problem where we need to find the time and location of maximum truck arrivals from video feeds collected by tens of thousands of geo-distributed cameras in a city. This is an expensive-to-evaluate function since it requires performing analytics on a huge volume of video feeds. It is natural to model the truck arrivals (over time and location) as a Gaussian Cox process. We can sample a small set of video clips and use them to estimate the latent intensity, thus supporting efficient BO. (2) Consider the problem of finding the time and location of maximum bird-related incidents in a city. Again, this is an expensive-to-evaluate function, and we have to estimate a latent intensity function from limited samples. We can use the proposed method to model the incidents through a Gaussian Cox process model and then support a BO to solve this problem. For problems requiring estimating a latent intensity function from point observations, we need to use BO with Gaussian Cox process models as proposed in this paper.
>
> >Reviewer: In the literature review, the authors write that "existing works mostly concentrate on the mean estimation". I am not an expert in Cox models, but it seems that this has been investigated in [1]. This means that this approach could have also been tested in the numerical experiments with Bayesian Optimization. It also seems that this cited paper uses some tools (Laplace approximation, eigenfunctions decomposition) similar to the ones in the paper. Would it be possible to highlight the main modeling differences with this paper? What is the theoretical/computational benefit of the paper's approach compared to the cited paper?
>
> ***Response***: Thanks for the question. In the paper, we have compared our results with those reported in [1]. Specifically, this is included in Table 2 with respect to three commonly used synthetic functions and the IQL metric. Regarding the approaches and modeling differences, we would like to clarify that the previous paper [1] uses path integral to obtain an approximation of the solution. In contrast, our approach is to recast the problem into a new RKHS through a change of kernel technique so it becomes easier to handle. It also allows applying the representer theorem to ensure the uniqueness of the estimated mean and covariance. Additionally, our approach enables the use of Nystrom approximation for efficient numerical computation (in Section 3.3), given the change of kernel and the new RKHS. These fundamental differences allow our proposed solution to achieve more favorable results than [1], as demonstrated in Table 2, and make our framework align more with the BO design.

---

> > ### Author Response · Authors · 2023-11-16
> > **Responses to Reviewer dgpT (2/2)**
> >
> > >Reviewer: In Equation (5), the authors claim that the rest of the terms of the likelihood are dominated by the first term when $n$ is large. However, this assumption is not always true in a BO setting (it is not in the numerical experiments). Could the authors comment that point and provide more details?
> >
> > ***Response***: This is a great question. This assumption of large $n$ is used by research on Gaussian Cox process models, such as [2], to make the problem tractable. Since $n$ denotes the number of point observations obtained, it accumulates and keeps increasing during the BO process. In practice, the value of $n$ climbs quickly and reaches a large value eventually so that this approximation will be more and more accurate after each step. We have also conducted additional experiments to evaluate the impact of such an approximation with a relatively small synthetic dataset. Based on the results in Fig. 10 in Appendix N, our framework can be used to predict the intensity, and the accuracy of the prediction will improve after several iterations.
> >
> > >Reviewer: The authors claim that they can use the Nystrom approximation to compute the next approximation from new samples in an incremental fashion. Is this step used in the experiment? How does it work in practice?
> >
> >
> > ***Response***: In the experiments, we utilized the Nystrom method, and the results are provided in the evaluation section, as shown by the mean and variance estimates in Fig.2. We note that this reduces the computational complexity to $O(n^2m+m^3)$ where $m \ll n$ [3]. To do this, first, we discretize the time span into a uniform grid. The pseudocode for this approximation is given by Algorithm 2 in the appendix. Once we have the approximated kernel matrix, we can compute the posterior mean and covariance for BO.
> >
> > >Reviewer: After reading the proof of Lemma 2, there is a point I did not understand. Why does the function $h$ belong to $\mathcal{H}$? This fact does not look straightforward to me. Perhaps I missed something in the development.
> >
> > ***Response***: RKHS is a functional space. Given a non-empty domain and a positive definite kernel $k$, we have a unique RKHS $\mathcal{H}_k$ of functions $h$. In $\mathcal{H}_k$, we can leverage the classical representer theorem to optimize $h$, as described in Section 3.2. By constructing a unique RKHS of $h$, we can utilize the property in $\mathcal{H}_k$ and derive a closed-form solution for posterior mean and covariance.
> >
> > >Reviewer: In the numerical experiments, the authors do not report the results of PIF in Table 2. Is there a reason for that? I've not found it in the paper.
> >
> > ***Response***: Since the PIF code in [1] has not been made publicly available due to some regulation reasons, we can only compare with the results based on numbers reported in Tabel S3 in [1], which only contains the IQL metric. Nevertheless, our method comfortably outperforms this baseline for the IQL comparison.
> >
> > >Reviewer: In figure 2c, it looks like the maximum of the acquisition function is around $t=40$. This means that we have to sample around $t=40$. However I don't see any sample around $t=40$ in figure 2d.
> >
> > ***Response***: Sorry about the confusion. The black vertical bars in Fig. 2 at the bottom of each subplot are the synthetic ground-truth events of the given ground-truth function. The red vertical bars beneath the black ones illustrate the cumulative observations till the current steps, representing the union of all observations from the initial step onward. Our BO algorithm is oblivious to ground-truth events (black vertical bars), which are shown for comparison with observations (red vertical bars). We state their definitions in Appendix G and will also point them out at the beginning of the experiment section.
> >
> > The green curve depicting the UCB function anticipates the region of interest for the next step based on our posterior estimates of the current step. Since there are no ground-truth events (black vertical bars) in this area, even though the acquisition function indicates $t=40$ as the next sampling region of step 15, no actual observations will be collected. Moreover, we provided the completed consecutive step-wise figure (Fig. 8 in Appendix L), demonstrating the BO procedure comprehensively.
> >
> > >**References** \
> > [1] Kim, Hideaki. “Fast Bayesian Inference for Gaussian Cox Processes via Path Integral Formulation.” \
> > [2] Schwarz, Gideon E, "Estimating the dimension of a model". \
> > [3] Baker, Christopher TH, and R. L. Taylor. "The numerical treatment of integral equations."
> >
> > > The code is made available via the [link](https://anonymous.4open.science/r/gaussian_cox_bo-6926).

---

> > > ### Comment · Reviewer_dgpT · 2023-11-22
> > > **Thanks for the responses**
> > >
> > > I thank the authors for the responses.
> > >
> > > I am still not convinced about an answer of the authors.
> > >  When the number of samples is low, I don't see any reasons to neglect the prior term in the likelihood function. Indeed, there are some practical cases where the number of acquired data points grows very quickly. However, there is also a large number of problems where acquiring data is costly and the purpose of using BO approach is then to make this acquisition intelligent with a low number of data points.
> > >
> > > I'd consider increasing my score.

---

> ### Author Response · Authors · 2023-11-23
> **Response to Reviewer dgpT**
>
> Thank you for your comment. We will certainly consider discussing the approximation related to a few samples and intelligent acquisition strategy with a low number of data points in future work. For now, we have added an experiment starting with a small number of events to observe each step in Fig. 10, Appendix N. The experimental results show that our method can be applied to cases where input events are low without significant estimation deviation from the true intensity.

---

### Official Review · Reviewer_bJz1 · 2023-10-27

**Soundness:** 3 good
**Presentation:** 2 fair
**Contribution:** 3 good
**Rating:** 8
**Confidence:** 4

**Summary:**

The paper presents a novel framework for conducting BO by leveraging Cox processes. This approach hinges on a Laplace approximation of the likelihood and uses kernel techniques to transform the optimization problem into a RKHS. The framework is empirically evaluated across a range of scenarios, encompassing well-known synthetic functions and real-world databases. The results of numerical experiments indicate that this approach exhibits competitive performance in comparison to other state-of-the-art methods. Unlike the other frameworks, it stands out by enabling BO within the context of Cox process-based models.

**Strengths:**

Theoretical contributions bring together techniques from the machine learning and functional analysis communities. Lemmas and other theoretical developments can be easily verified thanks to the clarity of the discussions. The diversity of examples, which take into account well-studied synthetic functions and real databases, makes it possible to assess the competitiveness of the framework in relation to the literature. The paper is generally well-written and well-organized.

**Weaknesses:**

Although the strengths lie in both the theoretical and numerical aspects, the motivation for performing BO in point processes lacks practical utility. For example, in the spatio-temporal application describing tornadoes in the USA, I fail to see how new events (tornadoes involving damage) can be sampled sequentially to promote active learning of the intensity function $\lambda$. I assume that for illustrative purposes, the authors considered adding the "closest event" available in the database that matches the BO's suggestion. Is this correct? If this is the case, and if the size of the database allows tractable implementations, we can consider all events for inference of $\lambda$. If the model cannot handle the whole database, the BO schema is an interesting idea that promotes a threshold between inference quality and the number of observed point events. However, what can be done if no similar events are recorded in the database?  Can the authors give further details on the practical utility of their framework?

The authors have suggested publishing the Python codes in a Github repository, but there is no evidence of their existence. I suggest sharing an anonymous repository (e.g. via https://anonymous.4open.science/) for further examination.

**Questions:**

**Questions**
- Are the results in Table 2 consistent, i.e. similar results are obtained for a different seed? If no, the authors must consider several random replicates and provide the mean +- std of the results
- In Figure 2, at the initial step, the UCB acquisition function suggests adding new events at $t > 90$ (since we seek to maximize such criterion) but they are added somewhere else. Similarly, in step 14, the UCB targets the instants around $t = 40$ but events are again added somewhere else. Besides the authors argue that "the algorithm keeps sampling by maximizing UCB acquisition function and then improving the estimation based on new samples observed", the plots do not validate their point. Can the authors further explain the results while clarifying my concern? Is it possible to add extra plots at consecutive steps (e.g. steps 1 and 2) for a better understanding of the BO's choice?
- In the experiments, the choice of the hyperparameters $w_1, w_2, w_3$ is not discussed. Can the authors precise their values in each experiment and explain how they were tuned?
- The authors approximate the integral $\int_{\mathcal{S}} \kappa(g(t)) dt$ using an $m$-partition Riemann sum to obtain a closed-form of the posterior covariance. Since such approximation depends on $m$, can the authors discuss the quality of the approximation in terms of $m$ and precise how they tune that value in the experimental setup? Can they also discuss the scalability of the approximation when $d$ increases?
- The limitations of the proposed framework are not discussed in the paper. Can the authors add a remark on this subject?

**Other minor remarks**
- Page 3, Table 1: the derivatives of the link functions need to be checked. For instance, $\dot{\kappa}(x) = 2x$ (quadratic case), $\dot{\kappa}(x) = \frac{e^{-x}}{(1+e^{-x})^2}$ (sigmoidal), $\ddot{\kappa}(x) = \frac{e^{-x}}{(1+e^{-x})^2}$ (softplus), ...
- Page 3, Section 3.1: $\Sigma$ is a **CO**variance
- Page 3, Section 3.1, after Eq. (2): $\lambda(t) = \kappa(g(t)) \to \lambda(t)$ (it has been already defined before Eq.(1) )
- Page 4, after Eq. (7): However, Equation equation (7)
- Page 4, after Eq. (8): $\eta_i$ and $\phi_i(\cdot)$ need to be defined in the main part of the paper (they were defined in the supplementary material)
- Page 5, Eq. (10): $\Lambda = \operatorname{diag}(\lambda_1, \ldots, \lambda_m)$ needs to be defined
- Page 7, Section 4: To precise that further details on the "evaluation setup" are given in Appendix G
- Page 7, Section 4.1.1: to indicate the number of events considered in each toy example
- Page 9, Figure 5: to indicate the iteration step in each panel
- In Appendix C, Eq. (26): $h(t_j) = \langle h, \tilde{k}(t_j, \cdot) \rangle_{\mathcal{H}_{\tilde{k}}}$ ($j$ rather than $i$)
- In Appendix C, Eq. (32): the first line must be $\sum_{i=1}^{n} \log(\kappa(g(t_i))) - \sum_{j=1}^m \kappa(g(t)) \Delta t$. Then, the sign of $\ddot{\kappa}^2(\hat{g}_i) \Delta$ must be inverted.
- In Appendix C, Eq. (32): given the proposed notation, it is not clear that the dimension of $\nabla_{\hat{g}}^{2} \Psi(\hat{g})$ matches the dimension of the $d \times d$ matrix $\Sigma$. Can the authors clarify this and/or propose a more readable notation?
- In the References: laplace $\to$ Laplace (Illian et al., 2012), bayesian $\to$ Bayesian (Kim, 2021), to add all the authors in (Lai et al., 1985), to complete the reference (Stanton et al., 2022), to be consistent with the names of the journals and conferences and the style of displaying them.

---

> ### Author Response · Authors · 2023-11-16
> **Responses to Reviewer bJz1 (1/2)**
>
> Thank you for pointing out and correcting the minor mistakes in our manuscript, which will definitely make the paper much more robust. We have updated our manuscript after checking all the remarks you kindly provided. Besides, we provide our responses regarding your concerns as follows.
>
> >Reviewer: The authors considered adding the "closest event" available in the database that matches the BO's suggestion. Is this correct? If this is the case, and if the size of the database allows tractable implementations, we can consider all events for inference of. If the model cannot handle the whole database, the BO schema is an interesting idea that promotes a threshold between inference quality and the number of observed point events. However, what can be done if no similar events are recorded in the database? Can the authors give further details on the practical utility of their framework?
>
> ***Response***:  Thank you for the question. We will certainly clarify our motivation in the final version. BO based on Gaussian Cox process models is useful when the target function (or dataset, as the reviewer mentioned) is expensive to evaluate.
>
> We will add two illustrative examples where the target function is hard to evaluate (either because it requires performing expensive analytics on a known dataset or due to the fact that obtaining the complete dataset is expensive). (1) Consider a problem where we need to find the time and location of maximum truck arrivals from video feeds collected by tens of thousands of geo-distributed cameras in a city. This is an expensive-to-evaluate function since it requires performing analytics on a huge volume of video feeds. It is natural to model the truck arrivals (over time and location) as a Gaussian Cox process. We can sample a small set of video clips and use them to estimate the latent intensity, thus supporting efficient BO. (2) Consider the problem of finding the time and location of maximum bird-related incidents in a city. Again, this is an expensive-to-evaluate function, and we have to estimate a latent intensity function from limited samples. We can use the proposed method to model the incidents through a Gaussian Cox process model and then support a BO to solve this problem.
>
> We also need to clarify that our proposed BO leverages Gaussian Cox process models to estimate the target function (or dataset as mentioned by the reviewer) from limited samples available. It is a Bayesian method and does not require similar data. The results support the development of acquisition functions that balance exploration (by collecting new samples) and exploitation (to optimize performance objectives) in BO.
>
> >Reviewer: Are the results in Table 2 consistent, i.e., similar results are obtained for a different seed? If no, the authors must consider several random replicates and provide the mean +- std of the results.
>
> ***Response***: Thank you for your suggestion. Table 4 was included in the appendix as the completed version of Table 2 with standard deviation. Due to the page limit, we shortened the table in the main text to save space. We will also add a description in the corresponding section for clarification and refer readers to the full version of Table 4 in the appendix.
>
> >Reviewer: In Figure 2, at the initial step, the UCB acquisition function suggests adding new events at $t>90$ (since we seek to maximize such criterion) but they are added somewhere else. Similarly, in step 14, the UCB targets the instants around $t=40$ but events are again added somewhere else. Can the authors further explain the results while clarifying my concern? Is it possible to add extra plots at consecutive steps (e.g. steps 1 and 2) for a better understanding of the BO's choice?
>
> ***Response***: Sorry about the confusion. In Fig. 2, the black vertical bars at the bottom of each subplot denote the ground-truth events that are not visible to BO. Only the red vertical bars are regions sampled already and thus visible in BO. For instance, at the initial step, our proposed algorithm tries to build a Gaussian Cox process model based on the observed events shown by red vertical bars ($t=25$ and $t=60$).
>
> The green curve depicting the UCB acquisition function anticipates the region of interest for the next step based on our posterior estimates of the current step. In Fig. 2(c), the maximum UCB acquisition value indicates that the algorithm plans to explore around $t=40$ in the upcoming step 15. However, since no events are observed near $t=40$ when we sample, no new observations (i.e., vertical red bars) are added at $t=40$. The Gaussian Cox process model is adjusted based on this new information, and the UCB acquisition function is updated accordingly to guide future samples/steps. We have added the complete consecutive step-wise figure (Fig. 8 in Appendix L, providing a comprehensive demonstration of the BO procedure.

---

> > ### Comment · Reviewer_bJz1 · 2023-11-20
> >
> > I thank the authors for addressing many of my concerns.
> >
> > > Response: Thank you for your suggestion. Table 4 was included in the appendix as the completed version of Table 2 with standard deviation. Due to the page limit, we shortened the table in the main text to save space. We will also add a description in the corresponding section for clarification and refer readers to the full version of Table 4 in the appendix.
> >
> > Is it possible to precise the number of replicates used to build Table 4? Can the standard deviations be considered when highlighting the best results? For instance, for $\lambda_1$, it is not very clear that the model **Ours (q)** outperforms the model **PIF (s)** due to the larger error intervals when comparing the $IQL_{.85}$ criterion. Similarly, for $\lambda_2$ and $IQL_{.50}$, both **Ours (q)** and **STVB** are competitive (I would say that the latter is better).
> >
> > Increasing the number of replicates may help to make error intervals finer.

---

> > > ### Author Response · Authors · 2023-11-21
> > > **Response to Reviewer bJz1**
> > >
> > > Thank you for your feedback. Currently, we use five replicates to compute the estimation results, and our comparisons primarily focus on the mean value following the existing work, such as [1]. For $\lambda_1$, **Ours (s)** has the best performance, while **Ours (q)** performs slightly better than **PIF(s)** regarding ${\rm IQL_{.85}}$ metric. For $\lambda_2$, regarding the ${\rm IQL_{.50}}$ mean and other metrics, **Ours (q)** is better than **STVB**. Although **STVB** has a near and slightly better estimation in several edge cases than ours on ${\rm IQL_{.50}}$, our method's overall performance is better and more stable. Therefore, this characteristic allows us to apply BO further on top of our estimation. To demonstrate our method more thoroughly, we conducted an additional experiment using ten and fifteen replicates. The results are provided in the following table. Generally, the error will be smaller if we have more replicates. We will also integrate the complete results into a new table in the final version.
> > >
> > > | Replicates | $\lambda_1, {\rm IQL}_{.85}$ | $\lambda_2, {\rm IQL}_{.50}$ |
> > > | :------ | :------: | :-----: |
> > > | Ours (q) 10 replicates | 7.34 (2.81) | 12.40 (1.10) |
> > > | Ours (q) 15 replicates | 7.31 (2.76) | 12.47 (0.98) |
> > >
> > > >**Reference**\
> > > >[1] Aglietti, Virginia, Edwin V. Bonilla, Theodoros Damoulas, and Sally Cripps. "Structured variational inference in continuous Cox process models."

---

> > > > ### Comment · Reviewer_bJz1 · 2023-11-22
> > > >
> > > > I thank the authors for this clarification. When judging the best model, I suggest considering the median rather than the mean if the focus is only on model accuracy (ignoring the dispersion associated with error intervals). The former statistic is known to be more robust. In my opinion, it's fairer to compare models on the basis of both accuracy and dispersion. This can achieved by considering boxplots. For instance, Table 4 (using 10 replicates) can be completed (or replaced) by boxplots (baselines vs error criterion) for each intensity function $\lambda_i$.

---

> > > > > ### Author Response · Authors · 2023-11-23
> > > > > **Response to Reviewer bJz1**
> > > > >
> > > > > Thank you for the precious suggestion. We did another experiment computing the median of our method and available baselines with ten replicates. The following table shows the median and error for the $l_2$-norm metric. Accordingly, we add a figure (Fig. 11, Appendix O) to visualize the tabular results into boxplots. We will take your advice and attach the complete visualization in the final version showing each metric regarding each intensity function $\lambda_i$.
> > > > >
> > > > > | $l_2$-norm | $\qquad\lambda_1$ | $\qquad\lambda_2$ | $\qquad\lambda_3$ |
> > > > > |:-------------|:-------------:|:-------------:|:-------------:|
> > > > > | Ours (q) | 2.73 (1.04) | 27.17 (6.76) | 2.81 (1.51) |
> > > > > | Ours (e) | 2.89 (1.03) | 32.13 (4.26) | 3.80 (0.84) |
> > > > > | Ours (s) | 2.70 (1.04) | 30.37 (7.14) | 3.79 (1.15) |
> > > > > | RKHS | 3.72 (1.60) | 44.28 (5.64) | 7.05 (2.03) |
> > > > > | MFVB | 3.22 (0.97) | 35.22 (7.15) | 4.05 (1.25) |
> > > > > | STVB | 2.91 (0.89) | 29.87 (4.01) | 4.49 (1.96) |

---

> > > > > > ### Comment · Reviewer_bJz1 · 2023-12-04
> > > > > >
> > > > > > I thank the authors once again for performing further experiments allowing a better understanding of the competitiveness of the framework. I have decided to increase the score to 8 (accept, good paper).

---

> ### Author Response · Authors · 2023-11-16
> **Responses to Reviewer bJz1 (2/2)**
>
> >Reviewer: In the experiments, the choice of the hyperparameters $\omega_1, \omega_2, \omega_3$ is not discussed. Can the authors precise their values in each experiment and explain how they were tuned?
>
> ***Response***: $\omega_1, \omega_2, \omega_3$ are weights for mean and covariance terms to balance their contributions in the acquisition function. When these values are small, BO will place higher emphasis on the mean, favoring exploitation in the process. In contrast, when they are large, BO favors exploration by placing higher emphasis on large variance. In our experiments, we maintain the fixed values for all these parameters, specifically as 0.8.
>
> Furthermore, additional experiments were conducted to assess the impact. In the experiments, we add two extra levels (0.6 and 1.0) for $\omega_3$ to compare with the original $\omega_3=0.8$ in Fig. 6 as an example. The results are shown in Fig. 9 in Appendix M, where if we increase the $\omega_3$ value, the acquisition function will encourage exploring areas with high uncertainty earlier.
>
> >Reviewer:  The authors approximate the integral $\int_\mathcal{S} \kappa(g(\boldsymbol{t})) d\boldsymbol{t}$ using an $m$-partition Riemann sum to obtain a closed-form of the posterior covariance. Since such approximation depends on $m$, can the authors discuss the quality of the approximation in terms of $m$ and precise how they tune that value in the experimental setup? Can they also discuss the scalability of the approximation when $d$ increases?
>
> ***Response***: Since the exact closed-form solutions are not tractable, we discretize the integral term for obtaining the computable closed-form covariance. In the experiment, we set $m$ as the grid size. Therefore, a higher $m$ leads to a finer granularity but higher computation complexity; otherwise a lower $m$ will increase the computational efficiency but with slightly lower quality. To explore the impact of this value, we conduct an additional experiment and show the results in the following table, where we used the synthetic intensity and ran BO for 10 steps. In the table, the runtime will increase with a higher $m$, while the performance does not deviate much.
>
> | $m$ | 200 | 400 | 600 | 800 |
> | :---  | :----: | :---: | :---: | :---: |
> | step 5 $l_2$-norm | 8.35 (3.1) | 10.54 (3.43) | 12.99 (2.78) | 11.91 (2.98) |
> | step 10 $l_2$-norm | 4.68 (1.84) | 5.38 (2.21) | 4.42 (1.02) | 4.08 (1.12) |
> | runtime | 6.03 (0.12) | 15.98 (0.23) | 34.81 (0.09) | 71.61 (0.20) |
>
> Regarding the scalability of the approximation, our solution computes the kernel matrix of each dimension separately and then applies the Kronecker product to construct the final kernel matrix as $K = K_1 \otimes \dots \otimes K_d$. For example, given $d=2$ where we have two kernel matrices with size $m_1$ and $m_2$, the time complexity will be $O(m_1^2 m_2^2)$.
>
> >Reviewer: The limitations of the proposed framework are not discussed in the paper. Can the authors add a remark on this subject?
>
> ***Response***: Thanks for the question. For future work, we will analyze more complex scenarios with the high-dimensional Gaussian Cox process model, and we will try to add some discussions about this in the final version.
>
> >The code is made available via the [link](https://anonymous.4open.science/r/gaussian_cox_bo-6926).

---

### Official Review · Reviewer_kMEd · 2023-10-31

**Soundness:** 4 excellent
**Presentation:** 3 good
**Contribution:** 3 good
**Rating:** 6
**Confidence:** 2

**Summary:**

In this work, the authors propose a novel method to estimate the posterior mean and covariance of the gaussian cox process model.

They do this by first approximating the posterior $p(g|{t_i})$ via Laplace approximation, and then using BIC to further simplify the computation. This is in terms of $\hat g$, which must be solved for by minimizing Eq. 6. To do this, they use RKHS along with a transformation of kernel to make the problem computationally cheap to solve. Once this is done, the posterior mean and covariance can be estimated by $\hat g$ and the expression in Eq. (9). For kernels that cannot be expanded explicitly, they also discretize and use a Nystrom approximation.

With a way of estimating posterior mean and covariance, one now is free to choose an acquisition function for the specific problem being solved. The authors discuss various settings in which different acquisition can be applied within this framework.

Experiments are carried out showing both the modelling of the latent intensity, as well as the full framework applied in various spatiotemporal settings.

**Strengths:**

This paper claims to be the first work on BO using Gaussian Cox Process models. I could not disprove this claim through a short search, and if true, I think shows a clear strength in its originality. Every claim seemed technically sound and I could not find any glaring problems, and there were a myriad of experiments demonstrating the method in various synthetic and real world settings. The results present are qualitatively and quantitatively compelling, and the whole paper is relatively clear to understand and well written.

**Weaknesses:**

Because many other people have not used Gaussian Cox Process models for BO before, I wonder how much modelling the latent intensity actually helps. I did not see any results or discussion on this, but it feels like a useful comparison to make to show that using GCP is actually more performant than standard BO.

**Questions:**

-I'm slightly confused about Section 3.4 in that it seems like one can choose any acquisition function that would solve their problem. What about using Gaussian Cox enables us to do this in contrast to standard BO?

-Were there any experiments done which could find the posterior mean and covariance in closed form (without Nystrom approximation)? I don't have good intuition for how much expressivity is lost in doing this approximation.

-The paper analyzes this method on spatial-temporal data, but couldn't I use this method with any temporal data?

---

> ### Author Response · Authors · 2023-11-16
> **Responses to Reviewer kMEd (1/2)**
>
> Thank you for your appreciation of our work. We provide our answers to your questions as follows.
>
> >Reviewer: Because many other people have not used Gaussian Cox Process models for BO before, I wonder how much modeling the latent intensity actually helps. I did not see any results or discussion on this, but it feels like a useful comparison to make to show that using GCP is actually more performant than standard BO.
>
>
> ***Response***: Thanks for your question. The Gaussian Cox process is the suitable model for addressing problems involving discrete point observations over time and space, such as arrivals, events, and occurrences. This process, characterized by a Poisson process modulated by a latent Gaussian process, proves essential for scenarios where the standard BO relying on a Gaussian prior falls short. The latter cannot effectively model expensive-to-evaluate functions in the context of point observation and spatio-temporal data. Thus, the estimation of latent intensity from discrete point observations becomes necessary, requiring doubly stochastic process models. Gaussian Cox process models are considered as the golden standard for spatio-temporal data [1,2]. This is what motivates our work.
>
> We will provide two illustrative examples demonstrating the unique advantages of using Gaussian Cox process models for BO. (1) Consider a problem where we need to find the time and location of maximum truck arrivals from video feeds collected by tens of thousands of geo-distributed cameras in a city. This is an expensive-to-evaluate function since it requires analyzing a huge volume of video feeds. It is natural to model the truck arrivals (over time and location) as a Gaussian Cox process, as proposed in this paper. We can sample a small set of video clips and use them to estimate the latent intensity, thus supporting efficient BO. (2) Consider the problem of finding the time and location of maximum bird-related incidents in a city. Again, this is an expensive-to-evaluate function, and we need to estimate a latent intensity function from limited samples. We can use the proposed method to model the incidents through a Gaussian Cox process model and then support a BO to solve this problem. For problems requiring estimating a latent intensity function from point observations, we need to use the Gaussian Cox process models proposed in this paper.
>
> >Reviewer: I'm slightly confused about Section 3.4 in that it seems like one can choose any acquisition function that would solve their problem. What about using Gaussian Cox enables us to do this in contrast to standard BO?
>
> ***Response***: As mentioned in the previous answer, for problems relating to discrete point observations over time and space, such as arrivals, events, and occurrences, they must be modeled as a (doubly stochastic) Gaussian Cox process. Gaussian process models in Standard BO are insufficient. In this paper, we first develop a Gaussian Cox process model to estimate the latent intensity from point observations and then, in Section 3.4, show that various acquisition functions can be developed in our proposed BO framework, supporting standard acquisition functions like UCB as well as more specialized acquisition functions defined on the latent intensity function, such as maximum idle time detection and change point detection, fully exploiting the temporal properties of the underlying data.
>
> >Reviewer: Were there any experiments done which could find the posterior mean and covariance in closed form (without Nystrom approximation)? I don't have a good intuition for how much expressivity is lost in doing this approximation.
>
> ***Response***:  This is a great question. We have conducted additional experiments to assess the approximation gap. It uses a synthetic set of point events and an RBF kernel, comparing the mean intensity estimations with and without Nystrom approximation. As shown in the following table, the approximation is quite accurate.
>
> | Trials    | 1 | 2 | 3 | average |
> | ----------- | :----------: | :-----------: | :-----------: | :-----------: |
> | $l_2$-norm | 0.79| 1.33 | 0.89 | 1.00 (0.23) |
>
> We would also like to note that Nystrom approximation can significantly reduce the computational complexity from $O(n^3)$ to $O(n^2m+m^3)$, where $m$ is the number of grid size and smaller than the number of events $n$ [3]. The theoretical analysis of the approximation gap has been discussed in [4]. Additionally, the Nystrom approximation proves beneficial when obtaining a closed-form solution for a specific kernel is not feasible through explicit Mercer’s expansion. We will cite these results and explain them in the final version.

---

> > ### Author Response · Authors · 2023-11-16
> > **Responses to Reviewer kMEd (2/2)**
> >
> > >Reviewer: The paper analyzes this method on spatial-temporal data, but couldn't I use this method with any temporal data?
> >
> > ***Response***: Yes. Considering temporal data is a special case of spatial-temporal data, our proposed method can be directly applied. In fact, Fig. 2 on page 8 of the paper shows an example of applying the proposed method to synthetic and real-world temporal data.
> >
> > >**References** \
> > [1] Cunningham, John P., Krishna V. Shenoy, and Maneesh Sahani. "Fast Gaussian process methods for point process intensity estimation." \
> > [2] Basu, Sankarshan, and Angelos Dassios. "A Cox process with log-normal intensity." \
> > [3] Baker, Christopher TH, and R. L. Taylor. "The numerical treatment of integral equations." \
> > [4] Zhang, Kai, Ivor W. Tsang, and James T. Kwok. "Improved Nyström low-rank approximation and error analysis."

---

> > > ### Comment · Reviewer_kMEd · 2023-11-22
> > >
> > > I thank the authors for their rebuttals. My questions were answered for the most part, and I think that the current score I have given is fair. The primary reason for this is that I am uncertain about the significance of this work to the BO community. It is, however, a sound and mathematically interesting paper.

---

> > > > ### Author Response · Authors · 2023-11-23
> > > > **Response to Reviewer kMEd**
> > > >
> > > > Thank you for your recognition. To summarize, as the two related contributions in this work, we build a Gaussian Cox process model for estimating the expensive-to-evaluate intensity functions relating to discrete point events over time/space, as well as design a BO framework that builds on such models for better exploiting the temporal/spatial characteristics of the data in a sample-efficient manner.

---

### Author Response · Authors · 2023-11-16
**General Responses to all Reviewers**

Thank you for your feedback! We have made corresponding revisions to the paper per the comments that reviewers kindly provided. The changes include addressing the typos, adding necessary clarifications, and attaching additional experimental results. In detail, the changes are:

* Fixing the typos on Pages 3, 4, and Appendix C
* Adding explanations for $\eta$, $\phi$, and $\Lambda$ on Page 4 and 5.
* Adding clarifications and experimental details on Page 7.
* Adding additional information for Fig. 5, Page 9.
* Removing the confusing notations and solving the issue in Appendix D.
* Solving the issues in Reference.
* Attaching additional experimental results in Appendix L, M, N, and O per reviewers' suggestions.

-- Authors of submission 6205

---

### Meta-Review · Area_Chair_tJK9 · 2023-12-06

**Metareview:**

The reviewers agree that the authors' approach to building a Cox process model with full uncertainty quantification over the intensity function and then performing Bayesian optimization with this model is an interesting and novel contribution. Ultimately, I think the most compelling piece of the paper is the Cox model inference procedure itself, which does seem novel to me and has some pretty clear potential advantages over existing formulations, including relatively recent work like the path integral formulation. Adding the additional experiments the authors performed in response to Reviewer bJz1 will be helpful.

**Justification For Why Not Higher Score:**

The experimental evaluation of the Cox process model itself is pretty reasonable, but while the authors include one real BO task, I think this aspect could have been made a bit more compelling. Frankly, other than the specific tasks the authors mention in the paper, I'm struggling to imagine a huge variety of applicable tasks at the intersection of BO and Gaussian Cox process models.

**Justification For Why Not Lower Score:**

The inference mechanism the authors' use for the Cox process model itself is really nice, and that aspect does seem pretty well evaluated. I think people might find the methods from this paper interesting for that reason alone, whether or not they are performing Bayesian optimization.

---

### Decision · Program_Chairs · 2024-01-16

Accept (poster)